

# A 14 year dataset of in situ glacier surface velocities for a tidewater and a land-terminating glacier in Livingston Island, Antarctica

Francisco Machío[1], Ricardo Rodríguez-Cielos[2], Francisco Navarro[3], Javier Lapazaran[3], Jaime Otero[3]

(1) Escuela Superior de Ingeniería y Tecnología, Universidad Internacional de La Rioja (UNIR), Calle Almansa, 101, 28040 Madrid, Spain.
(2) Departamento de Señales, Sistemas y Radiocomunicaciones, ETSI de Telecomunicación, Universidad Politécnica de Madrid, Av. Complutense, 30, 20040 Madrid, Spain.
(3) Departamento de Matemática Aplicada a las Tecnologías de la Información y las Comunicaciones, ETSI de Telecomunicación, Universidad Politécnica de Madrid, Av. Complutense, 30, 20040 Madrid, Spain.

*Correspondence to*: Francisco Machío (francisco.machio@unir.net)

## Abstract

We present a 14 year record of in situ glacier surface velocities determined by repeated GNSS measurements at a dense net of 52 stakes distributed across two glaciers, Johnsons (tidewater) and Hurd (land-terminating), located on Livingston Island, South Shetland Islands, Antarctica. The measurements cover the period 2000-2013 and were done at the beginning and end of each austral summer season. A second-degree polynomial approximation is calculated for each stake, which allows estimating the approximate velocities at intermediate times. This dataset can be useful as input data to numerical models of glacier dynamics, or for calibration and validation of remotely sensed velocities such as D-inSAR or SAR offset/coherence tracking velocities, for a region where very scarce in situ glacier surface velocity measurements are available.

**Link to the data repository:** http://doi.pangaea.de/ 10.1594/PANGAEA.846791.

## 1. Introduction

In situ measured glacier-surface velocities are an important source of information for glacier dynamics studies. The strain field is defined in terms of velocity gradients, and the stresses in terms of strains through the constitutive relationship (most often Nye's generalization of Glen's flaw law; e.g. Cuffey and Paterson, 2010, ch. 3). The velocity field gradients are thus responsible for observed deformation patterns such as e.g. folding or foliation, and damage expressions such as fracturing, faulting and crevassing (Hambrey and Lawson, 2000; Ximenis et al., 2000). Furthermore, observed surface velocities can give an insight on basal conditions. In particular, they have been used since long ago to infer basal drag (e.g. van der Veen and Whillans, 1989; Hooke et al., 1989).

Observed surface velocities are commonly used as input data to numerical models. Here, they could be directly used as Dirichlet boundary conditions at the glacier surface for the velocity field. However, most often traction-free boundary conditions (i.e. Neumann conditions) are set at the glacier surface, and the velocities are used instead for tuning model's free parameters such as the viscosity coefficient in the constitutive relationship or the basal drag coefficient in the sliding law. For long time, the numerical simulations considered such coefficients as constant over the entire glacier (e.g. Hanson, 1995; Martín et al., 2005; Otero et al., 2010). Recently, it is becoming more and more usual to establish the viscosity and/or the basal drag coefficients as functions of position. This is done by means of inversion procedures that heavily rely on observed velocities at the glacier surface. For instance, in the method introduced by Arthern and Gudmundsson (2010) and modified by Jay-Allemand et al. (2011), the surface velocities are used to solve the Dirichlet problem involved in the inverse Robin problem solving for the viscosity or basal drag coefficients. However, these inversion procedures require a large amount of measured velocities, which is seldom available from in situ measurements and thus recommend the use of remotely-sensed velocities, such as Interferometric SAR, SAR offset tracking or SAR coherence tracking velocities (e.g. Strozzi et al., 2002; Rignot and Kanagaratnam, 2006; Joughin et al., 2010; Wuite et al., 2015). But even in these cases in situ-measured glacier velocities are still of much interest, since they provide a means for calibration and validation of remotely-sensed velocities (e.g. Strozzi et al., 2008; Schellenberger et al., 2015). This is of interest in view of the recent efforts to derive time series for regional or global ice-velocity fields such as those involved in the MEaSUREs program (https://nsidc.org/data/nsidc-0484/versions/2, accessed on 07/05/2017), GoLIVE project (https://nsidc.org/data/golive, accessed on 07/05/2017) and ENVEO CryoPortal (http://cryoportal.enveo.at/, accessed on 07/05/2017).



In this paper, we present a 14 year record of in situ glacier surface velocities determined by repeated GNSS measurements at a dense net of stakes on two glaciers, Johnsons and Hurd, located on Livingston Island, South Shetland Islands (Fig. 1). These islands, located off the north-western tip of the Antarctic Peninsula, previously had a scarce record of in situ velocity observations, which included measurements in the late 1980s in Nelson Island (Ren Jiaven et al., 1995), earlier measurements in the late 1990s in Johnsons Glacier (Ximenis et al., 1999), and measurements in the Arctowski Icefield, the Bellingshausen Dome and the Central Dome of King George Island between 1999/2000 and 2008/09 (Blindow et al., 2010; Rückamp et al., 2010, 2011). Such in situ velocity measurements are critical for validation of the estimates of remote-sensor-based studies of ice discharge in the region such as those by Osmanoğlu et al. (2013, 2014) for King George and Livingston islands, respectively. An added interest of the presented velocity record is that it corresponds to both a tidewater glacier and a land-terminating glacier, two typical, but very different in dynamical behaviour, glacier types in this region.

## 2. Geographical setting

Our study area is Hurd Peninsula (62º 39-42' S, 60º 19-25' W), located in the south of Livingston Island, South Shetland Archipelago, Antarctica. This peninsula is the setting of Juan Carlos I Station (JCI), which provided the logistic support for our fieldwork (Fig. 1). Hurd Peninsula is covered by an ice cap that extends over an area of about 13.5 km$^2$ and spans an altitude range from sea level to about 370 m.a.s.l. It is partly surrounded by mountains that reach between 250 and 400 meters in height.

This ice cap can be divided into two main glacier systems. The first main unit is Johnsons Glacier, a tidewater glacier, mostly flowing north-westwards, which ends on a calving front of about 50 m in height, of which just a few meters (typically < 10 m) are submerged. This calving front extends approximately 500-600 m along the coast. The second main unit is Hurd glacier, flowing mostly south-westwards and terminating on land, with three main lobes named Sally Rocks (flowing south-westwards), Las Palmas (flowing westwards) and Argentina (flowing north-westwards). There are three additional smaller basins, all flowing eastwards to False Bay, which were excluded from this study because they are heavily crevassed icefalls which prevent safe field measurements.

The local ice divide separating Johnsons and Hurd lies between 250 and 330 m.a.s.l. (Fig. 1). Hurd Glacier has an average surface slope of about 3º, though the small westward flowing glacier tongues Argentina and Las Palmas are steeper, around 13º. Typical surface slopes for Johnsons Glacier range between 10º in its northern areas and 6º in the southern ones.

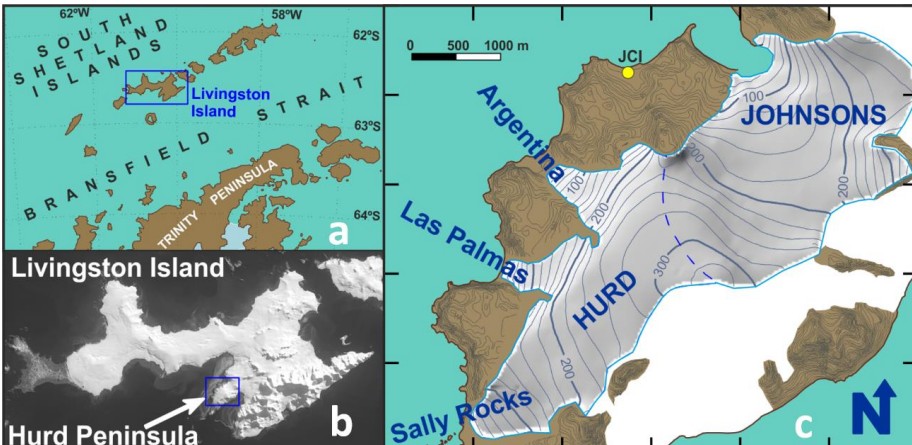

*Fig. 1. a) Location of Livingston Island in the South Shetland Archipelago. b) Situation of Hurd Peninsula on Livingston Island (ortophoto generated from SPOT 1991 image by Universitat de Barcelona and Institut Cartogràfic de Catalunya, 1992). c) Location and surface elevation map of Hurd and Johnsons Glaciers, Hurd Peninsula, Livingston Island. The dashed blue line indicates the ice divide separating Hurd and Johnsons Glaciers. Elevations and outline are based on a survey during summer 1998/99 and 2000/01. A yellow dot shows the position of Juan Carlos I Station (JCI).*



The Hurd Peninsula ice cap is a polythermal ice mass, showing an upper layer of cold ice, several tens of meters thick, in the ablation zone. This layer is rather uniformly distributed in Hurd Glacier, while it has a patchy distribution in Johnsons Glacier (Navarro et al., 2009). In the snouts of Hurd Glacier (in Sally Rocks area) and its side lobes Argentina and Las Palmas, where the glacier thickness tapers to zero, the cold ice layer extends down to bedrock, so the glacier is 5 frozen to the bed, implying a compressional stress regime. In contrast, the area close to Johnsons calving front shows the extensional stress regime characteristic of the terminus of tidewater glaciers (Molina et al., 2007; Navarro et al., 2009; Otero et al., 2010).

The average ice thickness of the joint Hurd-Johnsons, determined from ground-penetrating radar data in 2000/2001, was 93.6±2.5 m, with maximum values about 200 m, in the accumulation area of Hurd Glacier, and only about 160 m 10 in Johnsons Glacier (Navarro et al., 2009). Johnsons Glacier bed is rather regular, with altitudes decreasing towards the ice front, where glacier bed elevation is slightly below sea level (typically < 10 m). Hurd Glacier bed, however, is more irregular, with a clear over-deepening in the thickest ice area, close to the head of Argentina side lobe, and another one, though less pronounced, near the head of Las Palmas side lobe.

The Hurd Peninsula ice cap is subjected to the maritime climate of the western Antarctic Peninsula (AP) region. The 15 annual average temperature at JCI during the period 1994-2014 was −1.2ºC, with average summer (DJF) and winter (JJA) temperatures of 1.9ºC and −4.7ºC, respectively (Bañón and Vasallo, 2016). The surface mass balance over the period 2002-2011 has been close to zero for both glaciers: 0.05±0.30 mm w.e. for Johnsons and −0.15±0.44 mm w.e. for Hurd (the range indicates the standard deviation). The latter has shown a slightly more negative balance because of its lower accumulation rates, attributed to snow redistribution by wind, together with slightly higher ablation rates, due 20 to Hurd's hypsometry, which shows a much larger share of area at the lowermost altitudes (<100 m) as compared with Johnsons (Navarro et al., 2013). The average accumulation area ratios over the same period were 44±24 % for Hurd Glacier and 61±21 % for Johnsons Glacier (again, quoted the standard deviations). Their equilibrium line altitudes (ELA) for the same period were 228±57 m a.s.l. and 187±37 m a.s.l., respectively (Navarro et al., 2013).

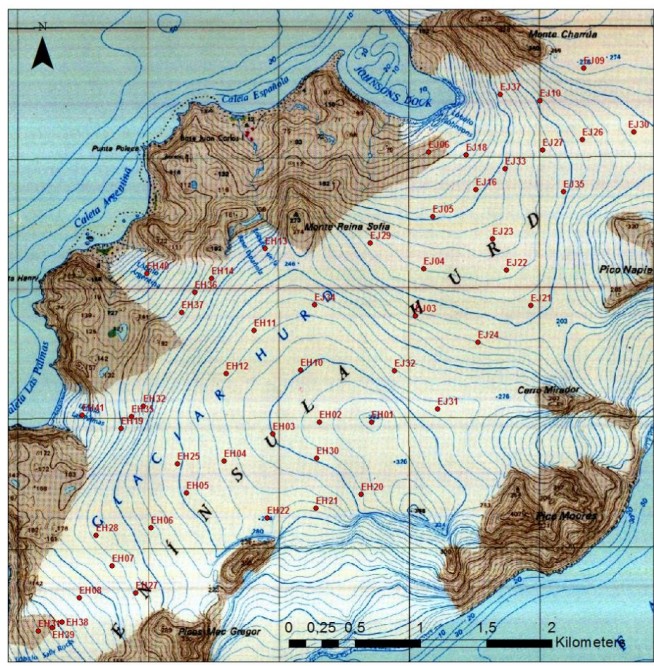

25

*Fig. 2. Net of stakes on Hurd and Johnsons Glaciers at the end of the 2012-2013 Antarctic summer campaign. (Base map: SGE, 1990).*





## 3. Methods

The glacier surface velocities were measured by precise repeated positioning measurements by GNSS techniques on a net of stakes deployed by the authors on Johnsons and Hurd glaciers (Fig. 2). The net of stakes consisted (as of the end of the measurement period reported) of 22 stakes for Johnsons and 30 stakes for Hurd Glacier. The stakes were surveyed 2-4 times per summer campaign during the period 2000-2013. The GNSS measurements were done using a Trimble 5700 system, with Data Controller TSC2. The measurements were done either in real-time kinematics (RTK) or in fast-static (post-processed) mode; for the latter, an occupation time of 10 s was set. In general, RTK mode was used, but in some cases a radio link to the base station was not available, and fast-static mode was employed. The GNSS base station was located at the neighbouring Juan Carlos I Station (Fig. 1). This is a permanent GNSS station with coordinates determined with an accuracy better than 5 cm in horizontal and 2 cm in vertical. The estimated horizontal accuracy for the stake positions lies between 0.07 and 0.60 m. The main contributor to this uncertainty is not the GNSS measurement error (average values of 7 and 10 cm for horizontal and vertical positioning, respectively) but the estimated uncertainties in the correction for tilt of the stakes.

From the collected observations of stake positioning, a surface velocity map can be obtained by applying the procedure described below. We will just focus on horizontal velocities, since the vertical component of the velocity is very small, and prone to errors such as those of tilt of the stake. From the known position $(x_{t_n}, y_{t_n})$ of a stake at a given time $t_n$ (expressed in days since the zero time for observations, $t_0$, which we arbitrarily set as 01/01/1999 at 00:00), with the subscript $n$ indicating the sequential number of the observation, we define:

$$\Delta x_{t_{n-1}}^{t_n} = x_{t_n} - x_{t_{n-1}}$$
$$\Delta y_{t_{n-1}}^{t_n} = y_{t_n} - y_{t_{n-1}}$$
$$\Delta X = \sum_{i=2}^{i=n} \Delta x_{t_{i-1}}^{t_i}$$
$$\Delta Y = \sum_{i=2}^{i=n} \Delta y_{t_{i-1}}^{t_i}$$

(1)

In this way, the planimetric position of a stake over time is defined by the discrete functions

$$X(t_n) = X(x_{t_1}, x_{t_1} + \Delta x_{t_1}^{t_2}, \dots, x_{t_1} + \Delta X)$$
$$Y(t_n) = Y(y_{t_1}, y_{t_1} + \Delta y_{t_1}^{t_2}, \dots, y_{t_1} + \Delta Y)$$

(2)

It is possible to adjust the previous functions by means of second-order polynomials, which is equivalent to assuming that the stake moves with constant acceleration:

$$X_a(t_n) = a_x t_n^2 + b_x t_n + c_x$$
$$Y_a(t_n) = a_y t_n^2 + b_y t_n + c_y$$

(3)

This set of two equations, with three unknowns each, will have a solution, or a better approximation to it, if sufficient observations ($n \geq 3$) are available for each stake.

The unknown coefficients are determined by least-square fitting, minimizing the residual vectors

$$\mathbf{R}_x = \begin{bmatrix} t_1^2 & t_1 & 1 \\ t_2^2 & t_2 & 1 \\ \dots & \dots & \dots \\ t_n^2 & t_n & 1 \end{bmatrix} \begin{bmatrix} a_x \\ b_x \\ c_x \end{bmatrix} - \begin{bmatrix} X(t_1) \\ X(t_2) \\ \dots \\ X(t_n) \end{bmatrix} = \mathbf{AC}_x - \mathbf{X}$$

$$\mathbf{R}_y = \begin{bmatrix} t_1^2 & t_1 & 1 \\ t_2^2 & t_2 & 1 \\ \dots & \dots & \dots \\ t_n^2 & t_n & 1 \end{bmatrix} \begin{bmatrix} a_y \\ b_y \\ c_y \end{bmatrix} - \begin{bmatrix} Y(t_1) \\ Y(t_2) \\ \dots \\ Y(t_n) \end{bmatrix} = \mathbf{AC}_y - \mathbf{Y}$$

(4)

By minimizing the above residuals for each of the existing stakes, we will get the adjusted functions, $X_a(t_n)$ and $Y_a(t_n)$, which allow to estimate how the position of each stake evolves with time.

The horizontal velocity of a stake will be given, from the time derivatives of the positions, by the expressions:

$$\mathbf{v} = v_x \mathbf{i} + v_y \mathbf{j}$$
$$v_x = X_a'(t_n) = 2 a_x t_n + b_x$$
$$v_y = Y_a'(t_n) = 2 a_y t_n + b_y$$
$$v_{xy} = \sqrt{v_x^2 + v_y^2}$$

(5)



To obtain the error estimates $e_x$ and $e_y$ (from which we calculate $e_{xy} = \sqrt{e_x^2 + e_y^2}$) of the adjusted functions, $X_a(t_n)$ and $Y_a(t_n)$, we follow the parametric adjustment procedure (see details in Ghilani, 2010), which has to be applied separately for $X$ and $Y$ (for brevity, we just describe it below for $X$). For a least-square fit, these equations are:

$$\mathbf{N} = [\mathbf{A}^T \mathbf{A}]$$
$$\mathbf{C}_x = \mathbf{N}^{-1}[\mathbf{A}^T \mathbf{X}]$$
$$\mathbf{R}_x = \mathbf{A}\,\mathbf{C}_x - \mathbf{X}$$
$$\widetilde{\mathbf{X}} = \mathbf{X} + \mathbf{R}_x$$
$$\mathbf{Q}_{xx} = \frac{1}{\sigma_{x0}^2}\,\mathbf{N}^{-1}$$
$$\mathbf{Q}_{\widetilde{x}\widetilde{x}} = \frac{1}{\sigma_{x0}^2}\,\mathbf{A}\,\mathbf{N}^{-1}\,\mathbf{A}^T$$
$$e_x = \sqrt{\frac{\mathbf{R}_x^T \mathbf{R}_x}{r}}$$

$\mathbf{X}$: Vector of observations
$\widetilde{\mathbf{X}}$: Vector of estimates
$\mathbf{A}$: Matrix of coefficients
$\mathbf{R}$: Vector of residuals
$\mathbf{C}_x$: Vector of unknowns (the coefficients in the polynomial adjustment)
$\mathbf{N}$: Cost or discrepancy matrix
$\mathbf{Q}_{xx}$: Observations cofactor matrix
$\mathbf{Q}_{\widetilde{x}\widetilde{x}}$: Estimates cofactor matrix
$e_x^2$: Reference variance
$r$: Number of degrees of freedom; $r = n - 3$, with $n$ the number of observations

(6)

The above equations, solved for each individual stake, assume that all stakes are given equal weight. From these equations, $\mathbf{C}_x$ is solved first to determine the coefficients of the second-degree polynomial adjustment. Then, the adjusted values $\mathbf{AC}_x$ are calculated and the residuals $\mathbf{R}_x$ computed, and finally the root-mean-square error in position $e_x$ is calculated. The process is repeated for the corresponding equations for the Y coordinate, to get the vector of polynomial adjustment coefficients $\mathbf{C}_y$ and the error estimate $e_y$.

We note that the above error estimates do not represent actual errors in the data points but an estimate of the average deviations (in a root-mean-square sense) of the data point positions with respect to their polynomial approximation.

To evaluate the relative error in velocity, say, for the $x$ component of the velocity, $e_{v_x}$, from the errors in position, $e_x$, and in timing, $e_t$, we use

$$\frac{e_{v_x}}{|v_x|} \cong \frac{e_x}{|\Delta x|} + \frac{e_t}{|\Delta t|} \tag{7}$$

Assuming that the error in timing is negligible as compared with the error in position, the above equation simplifies to

$$\frac{e_{v_x}}{|v_x|} \cong \frac{e_x}{|\Delta x|} \tag{8}$$

## 4. Description of the datasets

The shape file CNDA-ESP_SIMRAD_VELOCITY.shp available at PANGAEA database (http://doi.pangaea.de/10.1594/PANGAEA.846791) contains the position data for all stakes of Johnsons and Hurd glaciers for the period from 2000 to 2013. We describe below the contents of each individual field in the shape file.

- Field "t38_stake": The name of the stake under consideration (see stakes in Figure 2).
- Field "t38_t0": The zero time for the time variable. We set it as 01/01/1999 at 00:00 GMT.
- Field "t38_fecha": The date and time for the measurement, with "YYYYMMDDHHMMSS" format.
- Field "t38_inct": The period of time in days from "t38_t0" to "t38_fecha" ($t_n$ in the above equations).
- Field "t38_x": X coordinate in meters (UTM 20S) for the stake (considered in an ideal vertical position, after correction for tilt, if applicable) ($x_{t_n}$ in Equation 1).
- Field "t38_y": Y coordinate in meters (UTM 20S) for the stake (considered in an ideal vertical position, after correction for tilt, if applicable) ($y_{t_n}$ in Equation 1).
- Field "t38_x_ide": X coordinate in meters (UTM 20S) for the position of the stake for the given time, calculated using the second-degree polynomial adjustment ($X_a(t_n)$ in Equation 3).
- Field "t38_y_ide": Y coordinate in meters (UTM 20S) for the position of the stake for the given time, calculated using the second-degree polynomial adjustment ($Y_a(t_n)$ in Equation 3).
- Field "t38_vx": X component for horizontal velocity of the stake for the given time, expressed in meters per year, calculated from the second-degree polynomial adjustment ($v_x$ in Equation 5).




- Field "t38_vy": Y component for horizontal velocity of the stake for the given time, expressed in meters per year, calculated from the second-degree polynomial adjustment ($v_y$ in Equation 5).
- Field "t38_vxy": Absolute value of horizontal velocity of the stake for the given time, expressed in meters per year, calculated from the X and Y components of the velocity obtained from the second-degree polynomial adjustment ($v_{xy}$ in Equation 5).
- Field "t38_v_aci": Azimuth for horizontal velocity of the stake, expressed in sexagesimal degrees, at the date of the measurement.
- Field "t38_err_x": Root-mean-squared deviation for the X position of the stake, expressed in meters ($e_x$).
- Field "t38_err_y": Root-mean-squared deviation for the Y position of the stake, expressed in meters ($e_y$).
- Field "t38_max_x": Maximum error obtained for the X position of the stake, expressed in meters.
- Field "t38_max_y": Maximum error obtained for the Y position of the stake, expressed in meters.
- Field "t38_ax": The estimation for the "$a_x$" coefficient in the second-degree polynomial adjustment of the position X of the stake ($a_x$ in Equation 3).
- Field "t38_bx": The estimation for the "$b_x$" coefficient in the second-degree polynomial adjustment of the position X of the stake ($b_x$ in Equation 3).
- Field "t38_cx": The estimation for the "$c_x$" coefficient in the second-degree polynomial adjustment of the position X of the stake ($c_x$ in Equation 3).
- Field "t38_ay": The estimation for the "$a_y$" coefficient in the second-degree polynomial adjustment of the position Y of the stake ($a_y$ in Equation 3).
- Field "t38_by": The estimation for the "$b_y$" coefficient in the second-degree polynomial adjustment of the position Y of the stake ($b_y$ in Equation 3).
- Field "t38_cy": The estimation for the "$c_y$" coefficient in the second-degree polynomial adjustment of the position Y of the stake ($c_y$ in Equation 3).

## 5. Results

The procedure described in the Methods section was applied to every stake, producing the polynomial coefficients and the estimated horizontal positioning misfits shown in Table 1. To give an idea of the order of magnitude of the velocities and their associated errors, and of their spatial variations, we have included in table A.1 in the Appendix the calculated velocities for a given time.

| Stake | $a_x$ | $b_x$ | $c_x$ | $a_y$ | $b_y$ | $c_y$ | $e_{xy}(m)$ |
|---|---|---|---|---|---|---|---|
| EH01 | -0,0000000210 | -0,0007499061 | 634706,536 | -0,0000001022 | -0,0006149926 | 3046974,077 | ±0,41 |
| EH02 | -0,0000000260 | -0,0019490138 | 634314,937 | 0,0000000376 | -0,0004418479 | 3046972,371 | ±0,31 |
| EH03 | 0,0000000017 | -0,0024338880 | 633957,992 | 0,0000000387 | -0,0013300512 | 3046883,749 | ±0,25 |
| EH04 | 0,0000000208 | -0,0048559670 | 633602,733 | 0,0000000476 | -0,0016952866 | 3046680,172 | ±0,24 |
| EH05 | 0,0000000827 | -0,0061748145 | 633322,971 | 0,0000000176 | -0,0023909505 | 3046440,393 | ±0,15 |
| EH06 | 0,0000001329 | -0,0093454794 | 633066,469 | 0,0000001143 | -0,0041399749 | 3046183,071 | ±0,30 |
| EH07 | 0,0000002467 | -0,0103102794 | 632769,911 | 0,0000001116 | -0,0060335306 | 3045901,732 | ±0,32 |
| EH08 | 0,0000002202 | -0,0084839082 | 632511,087 | 0,0000002088 | -0,0073298445 | 3045661,923 | ±0,31 |
| EH10 | -0,0000000115 | -0,0021984650 | 634172,352 | 0,0000000196 | 0,0024960084 | 3047352,461 | ±0,18 |
| EH11 | -0,0000000060 | -0,0033251769 | 633822,848 | -0,0000000136 | 0,0040018668 | 3047646,355 | ±0,22 |
| EH12 | 0,0000000058 | -0,0029001077 | 633610,294 | 0,0000000058 | 0,0014133530 | 3047331,978 | ±0,50 |
| EH13 | 0,0000000989 | -0,0040996188 | 633908,319 | 0,0000000699 | 0,0034554549 | 3048276,555 | ±0,34 |
| EH14 | 0,0000003949 | -0,0066196722 | 633507,694 | -0,0000000176 | 0,0011105547 | 3048060,055 | ±0,73 |
| EH16 | 0,0000005378 | -0,0121620919 | 633315,315 | -0,0000003729 | 0,0088014677 | 3047778,378 | ±0,40 |
| EH18 | 0,0000005266 | -0,0079704077 | 632901,615 | 0,0000004036 | 0,0003691018 | 3047006,754 | ±0,45 |
| EH19 | 0,0000001954 | -0,0062025268 | 632821,167 | 0,0000000024 | 0,0013091532 | 3046916,708 | ±0,37 |
| EH20 | -0,0000000041 | -0,0041607075 | 634641,903 | 0,0000000207 | -0,0018151514 | 3046428,116 | ±0,36 |
| EH21 | 0,0000000702 | -0,0069464784 | 634311,036 | -0,0000013781 | -0,0125603214 | 3046415,468 | ±0,80 |
| EH22 | 0,0000000418 | -0,0019778722 | 633913,716 | 0,0000000351 | -0,0024914249 | 3046250,139 | ±0,29 |



| Stake | $a_x$ | $b_x$ | $c_x$ | $a_y$ | $b_y$ | $c_y$ | $e_{xy}(m)$ |
|---|---|---|---|---|---|---|---|
| EH23 | 0,0000000850 | -0,0074476651 | 633495,325 | -0,0000000048 | -0,0040643877 | 3046056,767 | ±0,29 |
| EH25 | 0,0000001059 | -0,0067918760 | 633252,533 | 0,0000000337 | -0,0005615131 | 3046656,096 | ±0,19 |
| EH26 | 0,0000000665 | -0,0061374875 | 633283,461 | 0,0000000429 | -0,0008986263 | 3046923,597 | ±0,27 |
| EH27 | 0,0000001937 | -0,0088012166 | 632945,263 | 0,0000000846 | -0,0044487636 | 3045685,828 | ±0,31 |
| EH28 | 0,0000003036 | -0,0065784553 | 632626,203 | 0,0000002354 | -0,0039697127 | 3046120,387 | ±0,35 |
| EH30 | -0,0000001863 | -0,0031881664 | 634304,297 | -0,0000004025 | -0,0036269810 | 3046722,181 | ±1,28 |
| EH31 | 0,0000004603 | -0,0077532371 | 632191,166 | 0,0000002463 | -0,0042862662 | 3045393,753 | ±0,49 |
| EH32 | 0,0000004228 | -0,0056874228 | 632981,252 | 0,0000000427 | 0,0003263696 | 3047088,876 | ±0,82 |
| EH34 | 0,0000028295 | -0,0221847247 | 633412,561 | -0,0000004808 | 0,0070931604 | 3047929,344 | ±0,27 |
| EH35 | -0,0000001494 | -0,0043610240 | 632899,866 | 0,0000001839 | 0,0002225731 | 3047005,252 | ±0,34 |
| EH36 | -0,0000000071 | -0,0044714224 | 633378,639 | -0,0000008964 | 0,0142598599 | 3047909,661 | ±1,35 |
| EH37 | 0,0000001050 | -0,0073407541 | 633291,187 | -0,0000008597 | 0,0132842232 | 3047763,133 | ±0,83 |
| EH38 | 0,0000000544 | -0,0065644463 | 632376,878 | -0,0000002724 | -0,0024698302 | 3045462,670 | ±0,32 |
| EH39 | -0,0000000968 | -0,0050881561 | 632296,779 | -0,0000001033 | -0,0031726807 | 3045423,513 | ±0,18 |
| EH40 | 0,0000001909 | -0,0080001652 | 633027,591 | -0,0000000767 | 0,0064788163 | 3048074,317 | ±0,09 |
| EH41 | 0,0000016791 | -0,0191464499 | 632550,165 | -0,0000027307 | 0,0262539089 | 3046960,654 | ±0,57 |
| EJ03r | -0,0000001531 | 0,0100735338 | 634980,227 | 0,0000004252 | 0,0182063987 | 3047658,850 | ±0,76 |
| EJ04 | -0,0000003723 | 0,0061601161 | 635075,122 | -0,0000005812 | 0,0259414549 | 3048020,871 | ±1,22 |
| EJ05 | 0,0000000492 | 0,0011547502 | 635161,004 | -0,0000003319 | 0,0333842531 | 3048375,872 | ±1,79 |
| EJ05r | -0,0000004208 | 0,0031517495 | 635155,477 | -0,0000019291 | 0,0363184602 | 3048344,319 | ±1,57 |
| EJ06 | -0,0000011073 | -0,0042784868 | 635185,239 | 0,0000026943 | 0,0365926573 | 3048770,902 | ±3,07 |
| EJ06r | 0,0000016211 | -0,0103198593 | 635192,433 | -0,0000178994 | 0,0932484671 | 3048692,082 | ±3,82 |
| EJ09 | 0,0000000101 | -0,0000640752 | 636317,040 | 0,0000000306 | -0,0003560897 | 3049669,857 | ±0,17 |
| EJ10 | -0,0000004307 | -0,0068225266 | 636026,014 | 0,0000000111 | -0,0054558209 | 3049450,521 | ±0,48 |
| EJ11 | -0,0000056619 | -0,0199142185 | 635701,041 | 0,0000047076 | -0,0040812151 | 3049278,770 | ±1,13 |
| EJ14 | -0,0000083181 | 0,0057260572 | 635350,340 | 0,0000190604 | -0,0112107159 | 3048898,259 | ±4,77 |
| EJ14r | 0,0000026146 | -0,0265096848 | 635395,318 | -0,0000095539 | 0,0757322689 | 3048785,930 | ±1,54 |
| EJ15 | -0,0000115713 | -0,0150161625 | 635587,960 | 0,0000177466 | -0,0141862506 | 3049134,762 | ±4,52 |
| EJ16 | -0,0000012393 | -0,0075342532 | 635564,261 | 0,0000014961 | 0,0227882933 | 3048586,798 | ±1,74 |
| EJ16r | 0,0000001769 | -0,0118134174 | 635579,144 | 0,0000003110 | 0,0244004790 | 3048564,554 | ±1,49 |
| EJ17 | -0,0000073534 | -0,0169745300 | 635820,867 | 0,0000070748 | -0,0119983441 | 3049058,204 | ±1,84 |
| EJ17r | -0,0000017878 | -0,0295492517 | 635853,941 | 0,0000041007 | -0,0069866948 | 3049052,604 | ±1,13 |
| EJ18 | -0,0000047503 | -0,0128379987 | 635611,869 | 0,0000072099 | 0,0059753470 | 3048787,784 | ±3,62 |
| EJ18r | -0,0000020714 | -0,0198100707 | 635635,496 | 0,0000032380 | 0,0163894918 | 3048764,951 | ±2,08 |
| EJ19r | -0,0000027156 | -0,0347723467 | 635509,766 | 0,0000039813 | 0,0474550026 | 3048954,395 | ±2,89 |
| EJ21 | -0,0000000445 | -0,0012778820 | 635920,791 | 0,0000000648 | 0,0020946021 | 3047848,965 | ±0,12 |
| EJ22 | 0,0000000237 | -0,0039724530 | 635745,947 | 0,0000000882 | 0,0088414448 | 3048083,628 | ±0,50 |
| EJ23 | -0,0000000349 | -0,0046353861 | 635644,992 | 0,0000001090 | 0,0167557974 | 3048276,292 | ±0,87 |
| EJ24 | -0,0000000521 | 0,0032607734 | 635493,978 | -0,0000001090 | 0,0152334223 | 3047502,873 | ±0,52 |
| EJ26 | -0,0000009008 | -0,0103905425 | 636381,563 | -0,0000001309 | -0,0059871063 | 3049160,852 | ±0,70 |
| EJ27 | -0,0000014544 | -0,0219763759 | 636156,950 | 0,0000004109 | -0,0107406457 | 3049090,100 | ±1,48 |
| EJ28 | -0,0000031887 | 0,0011132529 | 636126,270 | 0,0000006713 | 0,0002700461 | 3048619,696 | ±1,33 |
| EJ29 | -0,0000001806 | 0,0115365135 | 634635,956 | -0,0000000833 | 0,0097340264 | 3048288,181 | ±0,64 |
| EJ30 | 0,0000002245 | -0,0074398001 | 636727,973 | 0,0000000834 | -0,0051238313 | 3049205,806 | ±0,13 |
| EJ31 | -0,0000002608 | 0,0064071706 | 635177,240 | -0,0000004367 | 0,0119751394 | 3047018,091 | ±0,12 |



| Stake | $a_x$ | $b_x$ | $c_x$ | $a_y$ | $b_y$ | $c_y$ | $e_{xy}(m)$ |
|---|---|---|---|---|---|---|---|
| EJ32 | -0,0000002940 | 0,0083991531 | 634841,075 | -0,0000007905 | 0,0142317135 | 3047310,149 | ±0,27 |
| EJ33 | 0,0000040527 | -0,0808182396 | 636025,782 | 0,0000004695 | 0,0104154497 | 3048835,009 | ±0,67 |
| EJ34 | -0,0000001959 | 0,0026058368 | 634258,842 | -0,0000003753 | 0,0083361641 | 3047831,927 | ±0,09 |
| EJ35 | -0,0000030794 | 0,0145204365 | 636169,067 | -0,0000019491 | 0,0189886987 | 3048680,544 | ±1,49 |
| EJ36 | 0,0000098129 | -0,1574078351 | 636082,069 | -0,0000007515 | 0,0481336713 | 3048977,865 | ±0,99 |
| EJ37 | -0,0000076218 | 0,0437382981 | 635654,292 | 0,0000010401 | -0,0281496644 | 3049584,987 | ±2,83 |

*Table 1. Adjusted functions, $X_a(t_n)$ and $Y_a(t_n)$, for all the stakes of the glaciers under study. It also shows the root-mean-squared deviation $e_{xy}$ (in meters) made in the polynomial approximation of the position.*

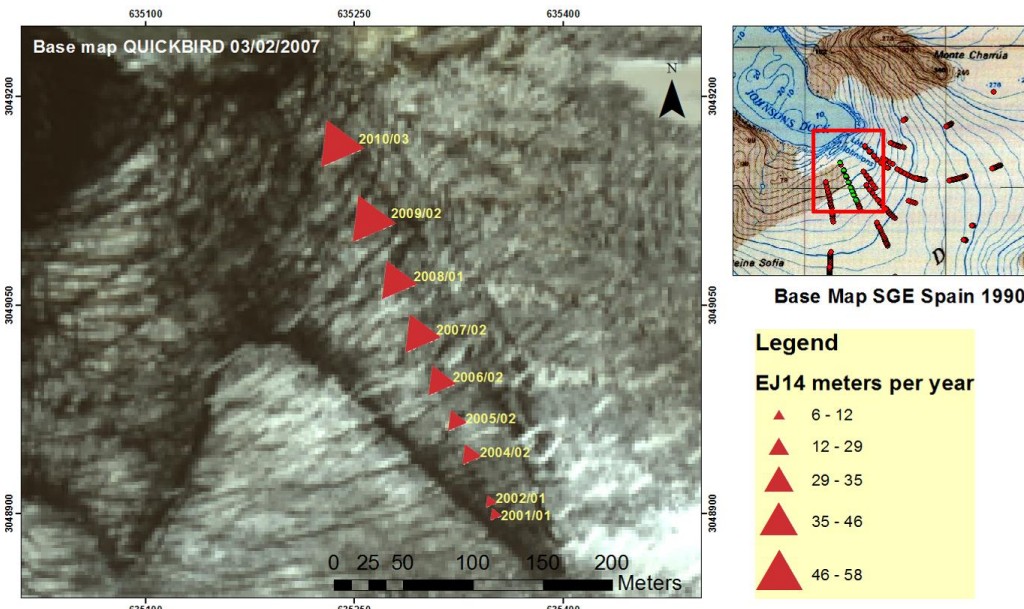

*Figure 3. Map showing the time evolution of Stake EJ14. Horizontal velocities and times for various positions are shown. The stake was lost by calving during 2010-2011. The inset to the right shows the location of the image shown to the left (in the inset, EJ14 trajectory is shown in green). In this, and the following figures, UTM coordinates (sheet 20S) are indicated. The background image is a satellite photo of the QUICKBIRD system program (2007).*

As an example, the detailed results for a particular stake, EJ14, are shown in Table 2 and Figure 3. The latter shows the position changes of the stake over time.

$$X_a(t_n) = -0.0000083181\, t_n{}^2 + 0.0057260572\, t_n + 635350.340$$
$$Y_a(t_n) = 0.0000190604\, t_n{}^2 - 0.0112107159\, t_n + 3048898.260$$
$$v_x = -0.0000166362\, t_n + 0.0057260572$$
$$v_y = 0.0000381208\, t_n - 0.0112107159$$
$$e_x = \pm 1.69\, m$$
$$e_y = \pm 4.46\, m$$
$$n = 25$$
$$Maximum\ velocity: 57.31\ m\ y^{-1}\ on\ March\ 1, 2010.$$
$$Maximum\ velocity\ azimuth: 336.7019°$$

*Table 2. Example of results for the adjustment by least squares of the position and the velocity of a stake (EJ14, near the calving front of Johnsons Glacier; see Figure 3), together with the root-mean squared deviations from the polynomial approximation for the position, as well as the maximum horizontal velocity and its direction.*



In Figures 4 and 5 we show the horizontal velocities for all stakes of Hurd and Johnsons glaciers, respectively, for a given date (13/02/2013), calculated using the corresponding polynomial adjustments. Maximum velocities on Hurd Glacier are only of a few meters per day, and approach 10 m y$^{-1}$ at the head of the unnamed glacier draining towards the south. Maximum velocities on Johnsons Glacier are much larger, up to several tens of meters per day, and reached near the calving front.

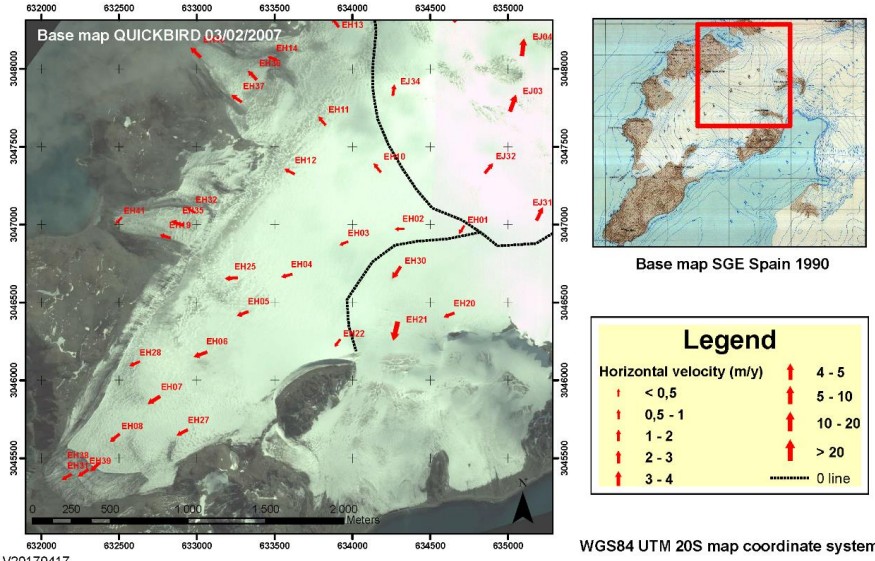

*Figure 4. Horizontal velocity (modulus and direction) for Hurd Glacier stakes, estimated for 13/02/2013 using a second-degree polynomial adjustment. The dotted lines indicate the position of the ice divides.*

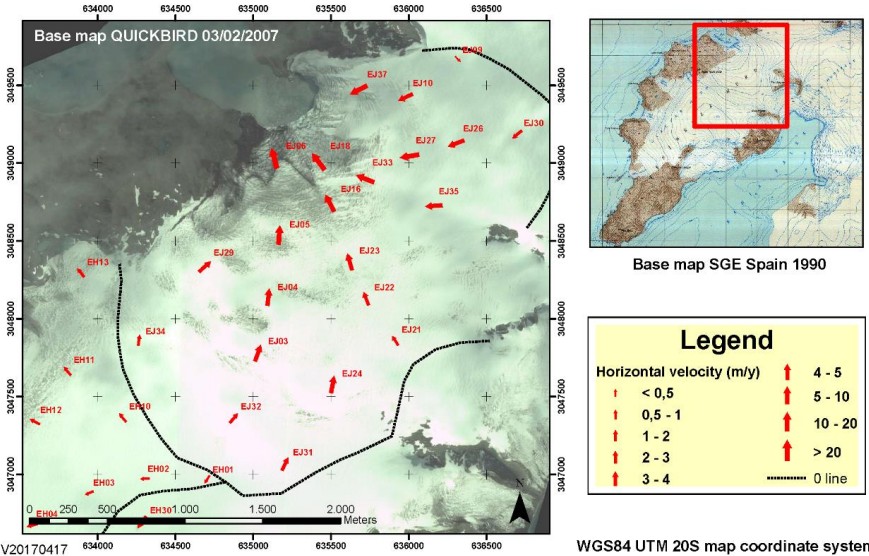

*Figure 5. Horizontal velocity (modulus and direction) for Johnsons Glacier stakes estimated for 13/02/2013 using a second-degree polynomial adjustment. The dotted lines indicate the position of the ice divides.*




In Figures 6 and 7 we show the corresponding spatially-interpolated (nearest neighbour) contour lines for the absolute value of the velocities for the same date. In these latter figures, the zero velocity bands indicate the approximate location of the ice divides, where the component of horizontal velocity normal to the divide plane is zero by definition, and the component tangent to that plane is usually very small.

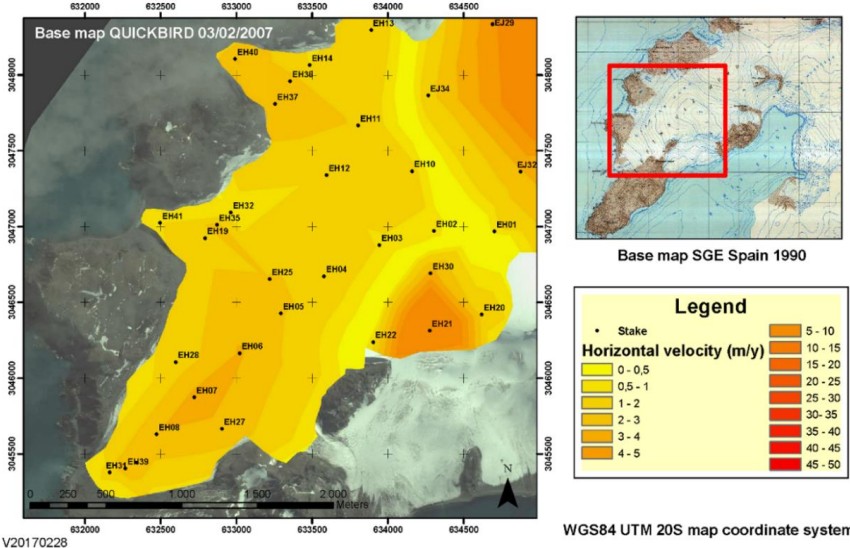

*Figure 6. Absolute value of horizontal velocity for Hurd Glacier, estimated for 13/02/2013 using a second-degree polynomial adjustment. The yellow near-zero velocity band indicates the approximate location of the ice divides.*

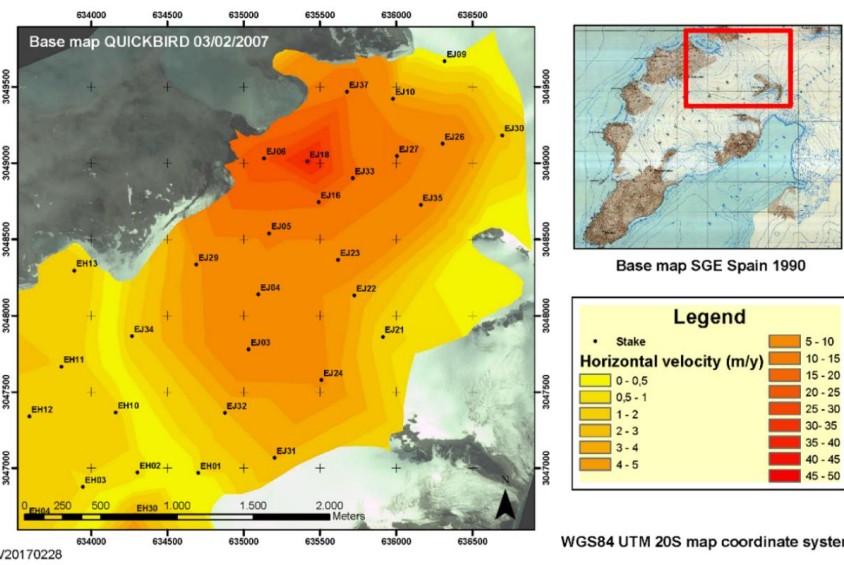

*Figure 7. Absolute value of horizontal velocity for Johnsons Glacier, estimated for 13/02/2013 using a second-degree*
10 *polynomial adjustment. The yellow near-zero velocity bands indicate the approximate location of the ice divides (except for the zone to the east, between UTM northing 3048000 and 3048500, which corresponds to thin frozen-to-bed ice on the upper part of a nunatak).*



## 5. Discussion and summarizing conclusions

From the analysis of Figures 4-7, we see that Johnsons and Hurd glaciers show two markedly different dynamical regimes. Since Johnsons is a tidewater glacier, it shows a pattern of velocities increasing from the ice divides (where horizontal velocities are zero by definition) towards its calving front, where yearly-averaged velocities up to 65 m y$^{-1}$ have been observed. On the contrary, Hurd is a land-terminating glacier, with much slower velocities (typically just a few m y$^{-1}$), in which the largest velocities are reached in its middle-to lower part (between stakes EH06-EH08; see Figure 6), where basal sliding is assumed to occur, and close to the land-terminating snouts the velocity field shows a decreasing pattern (this is particularly noticeable in the snouts of Sally Rocks and Las Palmas lobes; see Figure 6). Velocities are also high in the high-slope zones such as Argentina lobe and the upper part of Las Palmas lobe. Note that the high-velocity zone shown to the southeast of Hurd glacier, around stake EH21 (Fig. 6) does not really correspond to Hurd Glacier, but to an unnamed glacier flowing southwards, towards False Bay, which has extremely high slopes and is in fact a heavily crevassed icefall.

The decreasing velocities as we approach the land-terminating snouts have been attributed to the fact that the surficial cold ice layer reaches the bed in these zones, so the glacier is frozen to its bed and glacier movement is produced by internal deformation alone (no basal sliding). This is supported by both geomorphological observations, in particular the presence of compressional structures such as thrust faults close to the glacier termini (Molina et al., 2007; Molina, 2014) and to ground-penetrating radar studies that show that the cold ice layer reaches the bedrock in these zones (Navarro et al., 2009; Molina, 2014).

From the analysis of the polynomial interpolation of observed positions we see that a second-degree polynomial function (representing a uniformly accelerated motion) is sufficient to provide a fair adjustment to the observed position changes. The largest root-mean-square positioning error, of 5.54 m, is found for stake EJ18, which has an average horizontal velocity of ca. 30 m y$^{-1}$. Of course, one of the major drawbacks from the polynomial interpolation of the observed positions is that it does not allow to represent seasonal variations in glacier velocities, which are known to occur for the glaciers in this region (e.g. Osmanoğlu et al., 2014). In fact, we tried to add a sinusoidal function to the polynomial fit and the results were disappointing, although anticipated. This is because the positioning measurements are done only at the beginning and the end of each summer season, and thus do not allow to resolve yearly cycles. But the polynomial interpolation of all available positions for a given stake is just an example of what can be done with the available data. Calculations could be done for estimating e.g. summer-averaged velocities or winter-averaged velocities (for the "extended winter", all of the year except for the summer season). Yet, this is still insufficient to study velocity variations at scales shorter than the seasonal. For this reason, perhaps the highest interest of the presented dataset is its use for tuning of free parameters of numerical models of glacier dynamics (e.g. Martín et al., 2003; Otero et al., 2010), since these models represent averaged velocities at time-step scales, which are often of the order of weeks (especially for steady-state models such as those cited, in which the time steps are applied to get the model reach a steady-state configuration). But even for transient models weekly time steps are usual (e.g. Otero et al., 2017). The available dataset is also useful for validation of remotely-sensed SAR velocities, with typical repeat cycles from a few days to several tens of days, up to 45 days for ALOS PALSAR.

Another shortcoming of the presented dataset is that it does not allow for an easy analysis of dynamical response to climate changes (such as those regionally observed by Oliva et al., 2016), because what is available is a Lagrangian velocity field (velocities measured at stakes that change their position with time), while what is needed for studying glacier velocity variations in response to climate changes is an Eulerian velocity field (velocities measured at fixed location in space).

From the above discussion, a desirable complement to the available in situ velocity dataset presented in this paper would be a continuous record of ice velocities at selected stakes.

Summarizing, the presented dataset is a useful source of input data for numerical models of glacier dynamics and for calibration-validation of remotely-sensed velocity data. It fills an observational data gap in the region peripheral to the Antarctic Peninsula, and it is thus expected that these data will contribute to the understanding of the dynamics of the ice masses in this region and their response to environmental changes.

### Data availability

http://doi.pangaea.de/ 10.1594/PANGAEA.846791
Continuous velocity model for Johnsons and Hurd glaciers from 1999 to 2013, with link to model results in shapefile format.



**Acknowledgements**

This work was supported by grant CTM2014-56473-R from the Spanish National Plan of R&D.

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



Appendix A

| Stake | X velocity (m y$^{-1}$) | Y velocity (m y$^{-1}$) | Horiz. velocity (m y$^{-1}$) | Azimuth (°) | Error (m y$^{-1}$) |
|---|---|---|---|---|---|
| EH01 | -0,35 | -0,61 | 0,70 | 210,09 | ±0,83 |
| EH02 | -0,81 | -0,02 | 0,81 | 268,62 | ±0,90 |
| EH03 | -0,88 | -0,34 | 0,95 | 248,93 | ±0,63 |
| EH04 | -1,69 | -0,44 | 1,75 | 255,46 | ±0,60 |
| EH05 | -1,94 | -0,81 | 2,10 | 247,45 | ±0,35 |
| EH06 | -2,91 | -1,08 | 3,10 | 249,64 | ±0,57 |
| EH07 | -2,83 | -1,78 | 3,35 | 237,85 | ±0,55 |
| EH08 | -2,27 | -1,89 | 2,95 | 230,20 | ±0,38 |
| EH10 | -0,85 | 0,98 | 1,30 | 319,35 | ±0,39 |
| EH11 | -1,24 | 1,41 | 1,87 | 318,75 | ±0,69 |
| EH12 | -1,04 | 0,54 | 1,17 | 297,42 | ±1,31 |
| EH13 | -1,12 | 1,52 | 1,89 | 323,61 | ±0,89 |
| EH14 | -0,93 | 0,34 | 0,99 | 290,06 | ±1,34 |
| EH19 | -1,53 | 0,49 | 1,60 | 287,67 | ±1,12 |
| EH20 | -1,53 | -0,58 | 1,64 | 249,14 | ±0,64 |
| EH21 | -2,27 | -9,78 | 10,04 | 193,08 | ±0,71 |
| EH22 | -0,56 | -0,78 | 0,96 | 215,99 | ±0,88 |
| EH25 | -2,08 | -0,08 | 2,08 | 267,85 | ±0,61 |
| EH27 | -2,48 | -1,31 | 2,81 | 242,27 | ±0,80 |
| EH28 | -1,26 | -0,56 | 1,38 | 245,91 | ±0,73 |
| EH30 | -1,87 | -2,84 | 3,40 | 213,30 | ±4,15 |
| EH31 | -1,10 | -0,64 | 1,27 | 239,86 | ±0,96 |
| EH32 | -0,48 | 0,28 | 0,56 | 300,13 | ±2,76 |
| EH35 | -2,15 | 0,77 | 2,29 | 289,76 | ±1,02 |
| EH36 | -1,66 | 1,83 | 2,47 | 317,76 | ±2,10 |
| EH37 | -2,28 | 1,61 | 2,79 | 305,17 | ±1,30 |
| EH38 | -2,19 | -1,93 | 2,92 | 228,65 | ±0,31 |
| EH39 | -2,22 | -1,55 | 2,71 | 235,15 | ±0,27 |
| EH40 | -2,20 | 2,08 | 3,03 | 313,33 | ±0,09 |
| EH41 | -0,66 | -0,71 | 0,97 | 223,07 | ±0,49 |
| EJ03 | 2,42 | 6,57 | 7,00 | 20,25 | ±1,94 |
| EJ04 | 0,85 | 7,28 | 7,33 | 6,64 | ±1,28 |
| EJ05 | 0,61 | 10,94 | 10,95 | 3,17 | ±0,41 |
| EJ06 | -5,73 | 23,50 | 24,18 | 346,30 | ±1,72 |
| EJ09 | 0,01 | -0,01 | 0,02 | 135,41 | ±0,23 |
| EJ10 | -4,11 | -1,95 | 4,55 | 244,64 | ±0,58 |
| EJ16 | -7,41 | 13,95 | 15,80 | 332,01 | ±1,48 |
| EJ18 | -22,56 | 29,31 | 36,99 | 322,42 | ±5,54 |
| EJ21 | -0,63 | 1,01 | 1,19 | 327,85 | ±0,13 |
| EJ22 | -1,36 | 3,56 | 3,81 | 339,08 | ±0,72 |
| EJ23 | -1,82 | 6,53 | 6,78 | 344,39 | ±0,56 |
| EJ24 | 0,99 | 5,15 | 5,25 | 10,93 | ±0,56 |
| EJ26 | -7,18 | -2,68 | 7,67 | 249,55 | ±0,87 |



| EJ27 | -13,50 | -2,37 | 13,70 | 260,03 | ±2,80 |
| EJ29 | 3,53 | 3,24 | 4,79 | 47,47 | ±1,24 |
| EJ30 | -1,87 | -1,56 | 2,43 | 230,24 | ±0,15 |
| EJ31 | 1,36 | 2,73 | 3,05 | 26,45 | ±0,11 |
| EJ32 | 1,96 | 2,22 | 2,96 | 41,46 | ±0,65 |
| EJ33 | -14,23 | 5,57 | 15,28 | 291,38 | ±1,04 |
| EJ34 | 0,21 | 1,63 | 1,64 | 7,45 | ±0,11 |
| EJ35 | -6,29 | -0,41 | 6,31 | 266,30 | ±2,48 |

*Table A.1. Horizontal velocities for Hurd and Johnsons Glacier stakes on 13/02/2013, calculated using the second-degree polynomial adjustment. From left to right: stake name, X and Y components of horizontal velocity, absolute value of horizontal velocity, azimuth of the horizontal velocity vector, and error estimate for the horizontal velocity. All velocities are expressed in meters per year.*

