# Peer review of "A 14 year dataset of in situ glacier surface velocities for a tidewater and a land-terminating glacier in Livingston Island, Antarctica"

_Earth System Science Data, 2017_

## Referee Comment (RC1) · C. Völksen (Referee) · 19 Jun 2017

Dear Authors, it was my task to read your paper and to review it. I found your data set very interesting and valuable. I was an enormous effort to collect these data over 14 years in order to estimate the velocities from the positions. Nevertheless, I have some comments and suggestions, which I hope will improve the paper.

General comments:

=============

I would recommend to provide - with the data - an additional file (e.g. Readme) that

describes the content of the Zip-archive. As a non-GIS user I found it a bit complicated to access the data. After installing an open source GIS on my Linux PC I was able to read the data and to visualize them, which was helpful but took some time. Therefore, providing a simple ASCII file that contains the data of the shape file would be very useful. It makes the access to non-GIS users easier. Nevertheless, I have understood the advantages of a GIS environment.

The authors have provided a complete data set open to any potential users. The large advantage is also that every user can derive its own velocity field using a different approach. But in order to do so, the authors should add some additional information. For example, the provided data file contains additional stations, which are present in the available table and the appendix A. The reason is unknown.

Specific comments:

============

[P: page; L: Line]

P3/L16-20: Considering the standard deviations of the mass balance I would prefer a formulation like "... are not significant different from zero..". Therefore a discussion like " ... a slightly more negative balance because of" can be misleading based on the data. Can you add in a short sentence how the mass balances were estimated?

P4/top (3. Methods): In this context, you should speak about differential GNSS methods, since you used two GNSS-receiver: one as a base at JCI and one as the rover in the network of stakes. This should be clear for the reader from the beginning of this section. Please mention the distance between the base station and the rover. Since the base station operates permanently, I would have expected an accuracy better than 1 cm for the horizontal components. You should also be able to derive plate tectonics. Well, this does not matter so much for the local estimation of the velocity field.

Concerning the applied "fast static mode" with a site of 10 seconds only. I believe that

this is a very short occupation time. I wonder if you were able to resolve the ambiguities and estimate a precise position. Is it possible that the "fast static" sites do not fit well with the second-degree polynomial approximation? Can you comment on this fact?

I would also mention here that the estimated coordinates of the stakes were projected into the UTM-System using Zone 20S.

P4/L16: My first impression was that the variable t with the index n represents any possible time. While reading the paper, equation (2) and equation (4) imply that "n" stands for the last observation in time.

In this context, I also do not understand the purpose of equation (1). For the derivation of the second-degree polynomial approximation there is no need to form any delta x (or delta y etc.). To keep it clear I would simply remove it. It continues with equation (2), which in my eyes is rather the vector of observations (reduced to one component (e.g. X or Y)) containing all the positions of one stake for the entire period. Is it not the case that $X(t_n)$ (eq. 2) is the same as the vector X in equation (4)? I find this mathematical presentation confusing and recommend a simplified form.

Page 5 (top): You have decided to use a least squares approach treating all the observations with equal weight. On the previous page you write that the accuracy ranges between 0.07 to 0.6 m. I wonder now why you have decided to use equal weights. Would it not be possible to improve the results by using different weights? Can you comment on this?

Page 6 (equation (6)): The term Sigma_X0 is not explained. I suppose it is the a priori unit weight. In the same context I would consider e_x to be the "estimated variance of unit weight" a posteriori. In the next section you treat it as the root-mean-square error in position. I do not understand this expression and would rename it. The value of e_x rather indicates how well the approximation fits the positions of the stakes, as you say in the following sentences (P5/L11-L12). You could estimate the accuracy of the coefficients with (e_x**2)*inv(N). The elements on the diagonal of this matrix are the

**ESSDD**

variances of the coefficients.

Page (6) equation (7) and (8): I do not understand the origin of these two equations. My approach would be based on the following relation based on error propagation.

Velocity is given by two positions X2 and X1 and the time difference delta t [t2-t1] (assuming the error in time can be neglected):

V=(X2-X1)/(delta t)

Error propagation:

(S_v)**2= (-S_x1/(delta t))**2+(S_x2/delta t)**2

Assuming equal precision for the coordinates X1 and X2 gives:

Standard deviation of velocity: S_v= sqrt(2)/(delta t)*S_x => e_v=sqrt(2)/(delta t)*e_x

Therefore, I cannot see that the error of the velocity component is dependent on the size of the velocity or the distance between the two positions. It is dependent on the precision of the estimated positions and the time between the reoccupation.

Description of the datasets:

I found in the shape file four more fields addressed with dias, prevista_x, prevista_y and movxy. They are probably not necessary. But could you please describe their purpose or remove it? Could you also add information on the fields "t38_max_x" and "t38_may_y". How did you obtain the maximum error? I believe it is absolute value of the maximum residual (Rx/Ry).

I used the data of the shapefile and converted them into an ASCII readable text file. Based on that I re-computed some of your coefficients and velocities using Octave/Matlab. Testing it on EJ14 I was able to recover your results with small differences. Xa(tn)= -8.31294e-06*tn*tn+5.62765e-03*tn + 635350.468 Ya(tn)=1.90564e-05*tn*tn+-1.11635e-02*tn+3048898.201 Ex: 1.68 ey: 4.45 Vmax: 57.31 Azi: 336.68

But for some others like EJ35 I could not get the same results. Is it possible that you have removed outliers that were ignore during estimation? In the same context I noticed that some stations were called EJ14r or EJ14R. First, what is the difference between EJ14 and EJ14R? Is there a difference or are these stations identical? Also, I find in the list more stations than shown in the table (e.g. EH14, EH16, EH18, EH23, EH26 etc.). Is there a specific reason to remove those? It might be necessary to revise the table and explain the reason for removing those stations from the table.

Technical comments:

=============

Please consider most of my remarks as suggestions. I do not insist on a complete implementation. Since I myself do not speak English as a mother tongue, some corrections have to be treated carefully. [P: page; L: Line]

P1/L14: . . . repeated GNSS measurements in a dense network of 52 stakes ..

P1/L16: . . . 2000-2013 and were "performed/carried out" at the . . .

P1/L18: . . . . This dataset "is" useful as input . . .

P1/L20: D-InSAR (not D-inSAR)

P1/L24: . . . source of information for "the study of glacier dynamics".

P1/L31: . . . commonly used as input data "for" numerical models.

P1/L33: . . . are used instead for tuning "the" model's free . . .

P1/L36: . . . more and more "common" to establish ..

P1/L44: are still of "large" interest, since . . .

P1/L48: , "the" GoLive project . . . . . . and "the" ENVEO CyroPortal

P2/L2: . . .measurements "in" a dense "network" of stakes . . .

P2/L5: .. in the late 1990s "on" Johnsons . . .

P2/L26: .. (Fig. 1c)

P3/L26: Fig. 2. Network of stakes on . . .

P4/L2: The glacier surface velocities were estimated based on repeated differential GNSS measurements in a network of stakes deployed . . ..

P4/L3: . . . .The "network" of stakes consisted (as of the end of the reported measurement period) . . .

P4/L5. The GNSS measurements were "carried out/performed" using a Trimble 5700 system, with "d"ata "c"ontroller TSC2. The observations were "performed/carried out" either in ..

P4/L7: for the latter, an occupation time of 10 s"econds" was "used". . . .

P4/L9: e.g. The GNSS base station was located at the neighbouring Juan Carlos Station I (Fig.1 ) "in a distance of 2-5 km from the two glaciers".

P4/L14: From the collected positions of the stakes at different epochs, a surface velocity map . . ..

P4/L27: .. by "least squares fitting method", minimizing . . .

P5/L3: X). For a least squares "approximation, assuming observations of equal weight," these equations are:

P5/L6: This first sentence can be removed.

P6/Table1: Please define e_xy in the description of the table 1. It is not given in the text. I have checked it for EJ14 and there it is probably sqrt(ex*ex+ey*ey).

P8/ Figure 3: I find it very difficult to read the legend in the Quickbird map of this figure. Please make sure that the final print is showing the legend.

P9/ Figure 4 and 5: Same problem, I cannot read in the provided pdf-file the legend of the left map. The upper right map is not really necessary and also not readable. Both upper right maps of figure 4 and 5 show the same glacier (Johnsons).

P10/L2: . . . value of the velocity for the same date "(13/02/2013)". . . .

P10/Figure 6+7: Again, the legend is too small. I could imagine that a wider color spectrum makes it easier to identifier different zones. The here used spectrum from yellow to red makes it difficult to separate different zones.

P11/L5-10: Can you support your findings with numbers. Give values for the velocities and support terms like "decreasing pattern" or "high-velocity zone". What are the velocities for these?

P11/L11: . . ., which has extremely "steep" slopes. (?)

P11/L21: As mentioned before, I would not consider this as "root-mean-square position error".

P14/Appendix A: EJ14 and others are missing in the table! Also I would avoid the the error term in this table as long as it estimated based on formular (7) and (8),

---

## Referee Comment (RC2) · Anonymous Referee #2 · 5 Jul 2017

The authors present a 14-year dataset of in-situ measured surface velocities from an area in Antarctica that undergone a warming in the last decade(s). The dataset is very valuable for ice flow modeling or calibration of remotely sensed velocity fields (as already done by Otero et al., 2010; Osmanoglu et al., 2014). However, I have some comments that could help to improve the paper. As the other reviewer has already made many suggestions (that I have read, and I agree with most of them), I will try to avoid repeating suggestions.

COMMENTS

Abstract

[Figure]

I think, the abstract could state that velocities are only collected during summer season. So, possible seasonal variations are not covered.

Introduction

In general I think the Introduction could better state the provided dataset was already use for tuning an ice flow model (Otero et al., 2010) and calibration of remotely sensed velocities (Osmanoglu et al., 2014)

Page1, Line 32: It is correct, that observed velocities could be used as Dirichlet BC, but I would drop this sentence, because no ice sheet modeler apply that (to my knowledge).

Page 1, Line 34/37/40: What do you mean with viscosity coefficient? In the viscosity relationships the ice hardness, effective strain hardness, a power coefficient, the effective strain rate and the enhancement factor appear. To my knowledge, the enhancement factor or the ice hardness is tuned.

Page 1, Line 34/37/40: I prefer basal friction coefficient instead of basal drag coefficient. Geographical Setting

In Fig 2, I cannot identify the stake ID EJ14 (the one, that is plotted in Fig. 3). Also, a few more locations in Fig 2 are missing, if I compare the upper right inset in Fig. 3. The same holds for Fig. 5, it seems that some velocities at the marine terminating outlet of JG are not plotted (or not available?). I prefer to show every location in Fig. 2 and the following Figures. If in the result plots velocities are not available (for whatever reason), just show it with a black.

Methods

I think, the equations could be shortened by presenting only one component in each equation (similar as Eq. 6, 7, 8).

Page 4, Line 1-4: Can you explain the choice for the locations? Were you able to maintain the stakes for 14 years? So, none of the stakes was covered with snow or

fallen down due to melting? However, I am a bit confused, as I count more than 60 stake locations in the provided shape file.

Page 4, Line 5: Could you explain, why you take the measurements several times in each season?

Page 4, Line 6-13: Could you give a few more details about the positioning accuracy? How have you determined the tilt at each stake? And how do you get an error estimate from the tilt? As far as I understood, the positioning error does not enter the calculation of the velocity error (Eq. 7 and 8). The velocity error only depends on the polynomial interpolation. Is that right? The positioning error should also be provided in the data repository for each stake. Please use same units (m or cm) when specifying the accuracy. Just curious: In the user manual of the Trimble 5700 receiver (https://www.ngs.noaa.gov/corbin/class_description/5700-5800V2UserGuide.pdf; Page 92), I found accuracies for both modes (RTK and fast-static) that are depending on the baseline length. Have you used these formulas?

Page 4, Line 14: Please rewrite. The procedure described here does not create the surface velocity map (-> nearest neighbor interpolation, Page 10, Line 1), it describes the time interpolation. You should also somewhere motivate the polynomial interpolation. I think, it is a nice method, but I don't really see the benefit by using the interpolated velocities compared to the direct measured velocities (just extrapolate the measured displacements to meter per year).

Page 4, Eq. 3 and Table 1: Can you specify the units of the coefficients $a_x$, $b_x$ and $c_x$?

Page 5, Eq. 6: $\sigma_{x0}^2$ is not explained. In the legend the vector of residuals R is missing the subscript x.

Description of datasets

In the shape file I found 26 fields instead of the 22 fields described here. I also suggest

uploading the data for non-GIS users as a simple ascii file to the PANGAEA database. I think, the chapter could be moved to the Appendix.

Results

I recommend switching the Tables (Table 1 in the Appendix and the Table A1 in the main text), as the result of velocity calculation is the main task of this paper.

I plotted the data from the shape file in Qgis and have seen some strange behaviors (see attached figure). Most of the points are located on smooth lines/trajectories, but some outliers are observable. What is the reason for these outliers? Are these outliers considered in the velocity calculation or dropped?

In order to reduce the number of figures, I think Figure 4,5 could be overlaid on Figure 6,7.

Figures and tables

Figure 3: For colorblind people I recommend to use other colors than red and green.

Figure 4: The upper right inset marks JG instead of HG.

Figure 4,5: I suggest enlarging the arrows or plotting the velocity magnitude with different colors.

Figure 4-7: I cannot read the stake IDs. Please enlarge the stake IDs or provide a better figure quality (pdf version looks fine, but my printout not).

Table A1: The error provided is derived from Eq. 8?

For all tables, please move the caption to the top.
* * *
[Figure]

**Fig. 1.**

---

## Author Comment (AC1) · 2 Aug 2017

Dear Authors, it was my task to read your paper and to review it. I found your data set very interesting and valuable. I was an enormous effort to collect these data over 14 years in order to estimate the velocities from the positions. Nevertheless, I have some comments and suggestions, which I hope will improve the paper.

**General comments**
================

I would recommend to provide - with the data - an additional file (e.g. Readme) that describes the content of the Zip-archive. As a non-GIS user I found it a bit complicated to access the data. After installing an open source GIS on my Linux PC I was able to read the data and to visualize them, which was helpful but took some time. Therefore, providing a simple ASCII file that contains the data of the shape file would be very useful. It makes the access to non-GIS users easier. Nevertheless, I have understood the advantages of a GIS environment.

> *Following this suggestion, we have added a folder with the same information but in non-GIS users format (.xls and .txt), in addition to a text file including the description of the fields.*

The authors have provided a complete data set open to any potential users. The large advantage is also that every user can derive its own velocity field using a different approach. But in order to do so, the authors should add some additional information. For example, the provided data file contains additional stations, which are present in the available table and the appendix A. The reason is unknown.

> *Throughout the years some of the stakes have been lost (e.g. by iceberg calving at Johnsons Glacier front, or fallen because of intensive melting, or buried by heavy snowfalls) and new ones have also been added. This is why the PANGAEA dataset contains more stakes than those shown in the paper figures (which are snapshots in time). This has been clarified at the beginning of the new Methods section:*

*"We note that, over time, some of the stakes have been lost (e.g. by iceberg calving at Johnsons Glacier front, or fallen down because of intensive melting, or buried by heavy snowfalls) and new ones have also been added as replacement or to enlarge the original network. Because of this, there are differences in the set of stakes shown in the various figures in this paper, as they correspond to different snapshots in time. Also, the set of stakes included in the PANGAEA database (see Section 4) is larger than that in any of the figures, because it includes all of the stakes that have existed at any time within the complete measurement period."*

**Specific comments**
================

[P: page; L: Line]

**P3/L16-20**: Considering the standard deviations of the mass balance I would prefer a formulation like "... are not significant different from zero..." Therefore, a discussion like "... a slightly more negative balance because of" can be misleading based on the data.
Can you add in a short sentence how the mass balances were estimated?

*We have changed the first statement following your suggestion ("have not been significantly different from zero…"). However, we have remarked that the ranges indicated next to the averages are standard deviations (indicating a noticeable interannual variability), but that the actual estimated errors for each annual mass balance measurement is much lower, of the order of 0.1 m w.e. (by the way, there was a typo in the earlier version of the paper, and all values should be m w.e,, NOT mm w.e.). This means that it is fair (significant) to state that Hurd's balance is slightly more negative than Johnsons' balance, so we have kept this comment and its associated explanation.*

*We have also added, as suggested, a brief explanation on how the mass balances were measured: "Summer, winter and annual mass balances have been measured using the glaciological method on the same network of stakes used for the glacier velocity measurements, and then integrated to the entire glacier basins."*

**P4/top (3. Methods)**: In this context, you should speak about differential GNSS methods, since you used two GNSS-receiver: one as a base at JCI and one as the rover in the network of stakes. This should be clear for the reader from the beginning of this section.

*We have added the "differential" qualifier to make this more evident since the beginning of the section.*

Please mention the distance between the base station and the rover.

*The distance was between 2 and 4 km, depending on the stake position. We have added this in the new version of the manuscript.*

Since the base station operates permanently, I would have expected an accuracy better than 1 cm for the horizontal components.

You should also be able to derive plate tectonics. Well, this does not matter so much for the local estimation of the velocity field.

> *Sorry, this was a typo (the previously given data made reference to typical accuracies of our differential GNSS measurements at the stakes (rover). Indeed the error in the base station coordinates is much lower: 0.007 m in horizontal and 0.012 m in vertical. We have corrected this and added a reference for it (Ramírez-Rodríguez, 2007).*

Concerning the applied "fast static mode" with a site of 10 seconds only. I believe that this is a very short occupation time. I wonder if you were able to resolve the ambiguities and estimate a precise position. Is it possible that the "fast static" sites do not fit well with the second-degree polynomial approximation? Can you comment on this fact?

> *Thanks for pointing this out, as in fact there was a typo in this part of the text: the occupation time of 10 seconds actually corresponded to RTK measurements, while for fast-static measurements it was of 3-5 minutes, depending on the number of available (visible) satellites. It now reads as follows:*
>
> *"The measurements were performed either in real-time kinematics (RTK) or in fast-static (post-processed) mode; for the former, an occupation time of 10 s was set, and for the latter it was of 3-5 minutes depending on the number of satellites available."*

I would also mention here that the estimated coordinates of the stakes were projected into the UTM-System using Zone 20S.

> *We agree and have included it into the modified text.*

**P4/L16**: My first impression was that the variable t with the index n represents any possible time. While reading the paper, equation (2) and equation (4) imply that "n" stands for the last observation in time.

> *You are right. In fact, we recognize that the notation was confusing, so we have replaced the subscript "n" by "i" to make it clearer (and kept n as total number of measurements).*

In this context, I also do not understand the purpose of equation (1). For the derivation of the second-degree polynomial approximation there is no need to form any delta x (or delta y etc.). To keep it clear I would simply remove it. It continues with equation (2), which in my eyes is rather the vector of observations (reduced to one component (e.g. X or Y)) containing all the positions of one stake for the entire period. Is it not the case that X(t_n) (eq. 2) is the same as the vector X in equation (4)? I find this mathematical presentation confusing and recommend a simplified form.

> *You are right, and we have removed former Eq. (1) and changed the writing of the new Eq. (1) (formerly Eq. 2) to make it more understandable. Together with the mentioned change in subscripts and the rewriting of new Eq. (5) (formerly Eq. 6), we believe that the entire set of equations is now more understandable.*

**Page 5 (top)**: You have decided to use a least squares approach treating all the observations with equal weight. On the previous page you write that the accuracy ranges between 0.07 to 0.6 m. I wonder now why you have decided to use equal weights. Would it not be possible to improve the results by using different weights? Can you comment on this?

*Yes, it is right that the position accuracies for individual stake positioning measurements range between 0.07 to 0.6 m, but this does not mean that a particular stake shows systematically positioning accuracies in the low or the high end of the range. In practice, most stakes, over their histories, show both large and small accuracies, mostly depending on the tilt that they undergo at particular times. Tilt in the ablation zone (due to intensive melting) is in general more common than in the accumulation zone, but in the latter also many stakes undergo tilt because of strong winds (being anchored in softer snow/firn as compared to stronger ice in the ablation zone). In fact, we did not find a reasonable systematic way to assign different weights to different stakes, so we preferred to weight all of them equally.*

**Page 6 (equation (6))**: The term Sigma_X0 is not explained. I suppose it is the a priori unit weight. In the same context I would consider e_x to be the "estimated variance of unit weight" a posteriori. In the next section you treat it as the root-mean-square error in position. I do not understand this expression and would rename it. The value of e_x rather indicates how well the approximation fits the positions of the stakes, as you say in the following sentences (P5/L11-L12). You could estimate the accuracy of the coefficients with (e_x**2)*inv(N). The elements on the diagonal of this matrix are the variances of the coefficients.

*This comment has been addressed in the fully new writing of the equations and the associated text.*

**Page (6) equation (7) and (8)**: I do not understand the origin of these two equations. My approach would be based on the following relation based on error propagation.

Velocity is given by two positions X2 and X1 and the time difference delta t [t2-t1] (assuming the error in time can be neglected):
V=(X2-X1)/(delta t)

Error propagation:
(S_v)**2= (-S_x1/(delta t))**2+(S_x2/delta t)**2

Assuming equal precision for the coordinates X1 and X2 gives:
Standard deviation of velocity: S_v= sqrt(2)/(delta t)*S_x => e_v=sqrt(2)/(delta t)*e_x

Therefore, I cannot see that the error of the velocity component is dependent on the size of the velocity or the distance between the two positions. It is dependent on the precision of the estimated positions and the time between the reoccupation.

*We acknowledge this comment, as indeed the mentioned equations were improper. The new equations (6) and (7), and their associated text, clarify now the distinction between the interval velocity calculations (and associated error estimates) that the reviewer is*

*indicating and the distinct error estimate associated to the polynomial function describing the velocity, derived from the polynomial approximation for the positions of each stake.*

**Description of the datasets:**

I found in the shape file four more fields addressed with dias, prevista_x, prevista_y and movxy. They are probably not necessary. But could you please describe their purpose or remove it? Could you also add information on the fields "t38_max_x" and "t38_may_y".

*The information corresponding to these fields has been added both in the description made in the manuscript (now moved to Appendix A) and in the files available in PANGAEA.*

How did you obtain the maximum error? I believe it is absolute value of the maximum residual (Rx/Ry).

*The maximum error is the maximum residual of the polynomial interpolation (i.e. the maximum difference between the observed position and its corresponding polynomial approximation). This is included in the PANGAEA dataset, but not in the paper text.*

I used the data of the shapefile and converted them into an ASCII readable text file. Based on that I re-computed some of your coefficients and velocities using Octave/Matlab. Testing it on EJ14 I was able to recover your results with small differences.

Xa(tn)= -8.31294e-06*tn*tn+5.62765e-03*tn + 635350.468
Ya(tn)=1.90564e-05*tn*tn+-1.11635e-02*tn+3048898.201
Ex: 1.68
ey: 4.45
Vmax: 57.31
Azi: 336.68

But for some others like EJ35 I could not get the same results. Is it possible that you have removed outliers that were ignore during estimation?

*We did not remove outliers for EJ35, so we are not certain on the reason for the apparent discrepancy (we have redone our calculations and obtained our same earlier result). EJ35 has few observations (just 7) and this results in differences in the least significant digits. This could be a reason for the differences, but it is just a guess. On the other hand, EJ35 is a stake that shows an anomalous trajectory.*

In the same context I noticed that some stations were called EJ14r or EJ14R. First, what is the difference between EJ14 and EJ14R? Is there a difference or are these stations identical?

*This is a fully different problem. Stakes with and without the r (or R) suffix are totally different (though spatially close to each other). The short version is that, at a certain point in history some stakes were installed at exactly the same position that was occupied some years earlier by another stake (if the original stake was named e.g. EJ14, the new stake installed in its former position was named EJ14r).*

*The longer version is as follows: an earlier network of stakes, with fewer stakes, was deployed in the late 1990s in Johnsons Glacier –none in Hurd Glacier– by glaciologists from the University of Barcelona; they are the ones who did this experiment of installing new stakes in the former positions of another one, and in fact existed e.g. EJ14, EJ14r, EJ14rr, … This was an attempt to do some kind of "Eulerian" measurement of velocities (at fixed locations) versus the "Lagrangian" way of measurement, in which one follows the stakes as they move through the glacier. However, this procedure revealed to be too cumbersome and time- and resource-demanding. When we "inherited" in 2000/01 this network of stakes, we deployed additional stakes on Johnsons Glacier and deployed a new network on Hurd Glaciers, and we discontinued this practice (of r, rr, … stakes) from our colleagues, though kept the "r" stakes until they were lost (this took many years, and some are currently still "alive").*

*Of course, we should not explain this story in the paper text, but we have remarked on the stake list in PANGAEA dataset that stakes such as EJ14 and EJ14r, etc. are different.*

Also, I find in the list more stations than shown in the table (e.g. EH14, EH16, EH18, EH23, EH26 etc.). Is there a specific reason to remove those? It might be necessary to revise the table and explain the reason for removing those stations from the table.

*This has already been explained earlier in this "Answers to reviewer" file (and included in the new version of the manuscript).*

**Technical comments:**
=============

Please consider most of my remarks as suggestions. I do not insist on a complete implementation. Since I myself do not speak English as a mother tongue, some corrections have to be treated carefully.

[P: page; L: Line]

P1/L14: ...repeated GNSS measurements in a dense network of 52 stakes...

*Done.*

P1/L16: ...2000-2013 and were "performed/carried out" at the...

*Done.*

P1/L18: ...This dataset "is" useful as input...

*Done.*

P1/L20: D-InSAR (not D-inSAR)

*Done.*

P1/L24: ...source of information for "the study of glacier dynamics".

*Done.*

P1/L31: ... commonly used as input data "for" numerical models.

*Done.*

P1/L33: ...are used instead for tuning "the" model's free...

*Done.*

P1/L36: ...more and more "common" to establish .

*Done.*

P1/L44: are still of "large" interest, since...

*Changed to "wide interest".*

P1/L48: , "the" GoLive project ... and "the" ENVEO CyroPortal

*Done.*

P2/L2: ...measurements "in" a dense "network" of stakes...

*Done.*

P2/L5: ...in the late 1990s "on" Johnsons...

*Done.*

P2/L26: ... (Fig. 1c)

*Done.*

P3/L26: Fig. 2. Network of stakes on...

*Done.*

P4/L2: The glacier surface velocities were estimated based on repeated differential GNSS measurements in a network of stakes deployed...

*Done.*

P4/L3: ...The "network" of stakes consisted (as of the end of the reported measurement period) ...

*Done.*

P4/L5. The GNSS measurements were "carried out/performed" using a Trimble 5700 system, with "d"ata "c"ontroller TSC2. The observations were "performed/carried out" either in ..

*Done.*

P4/L7: for the latter, an occupation time of 10 s"econds" was "used" ...

*Done.*

P4/L9: e.g. The GNSS base station was located at the neighbouring Juan Carlos Station I (Fig.1 ) "in a distance of 2-5 km from the two glaciers".

*As we have previously indicated, this clarification has been added to the text.*

P4/L14: From the collected positions of the stakes at different epochs, a surface velocity map …

*Done.*

P4/L27: … by "least squares fitting method", minimizing …

*Changed to "…. by the least-square fitting method, minimizing …"*

P5/L3: X). For a least squares "approximation, assuming observations of equal weight," these equations are:

*Done.*

P5/L6: This first sentence can be removed.

*Done.*

P6/Table1: Please define e_xy in the description of the table 1. It is not given in the text. I have checked it for EJ14 and there it is probably sqrt(ex*ex+ey*ey).

*Your interpretation is correct, but this was already included in P5/L1 of the former text.*

**P8/ Figure 3**: I find it very difficult to read the legend in the Quickbird map of this figure. Please make sure that the final print is showing the legend.

*Done*

**P9/ Figure 4 and 5**: Same problem, I cannot read in the provided pdf-file the legend of the left map. The upper right map is not really necessary and also not readable. Both upper right maps of figure 4 and 5 show the same glacier (Johnsons

*Legends (and other aspects) of these figures have been improved. Furthermore, former figures 4 and 6, and 5 and 7, have been combined into a single figure (each pair) at the suggestion of the other reviewer.*

**P10/L2**: …value of the velocity for the same date "(13/02/2013)" …

*Done*

**P10/Figure 6+7**: Again, the legend is too small. I could imagine that a wider color spectrum makes it easier to identifier different zones. The here used spectrum from yellow to red makes it difficult to separate different zones. ).

*Done*

**P11/L5-10**: Can you support your findings with numbers. Give values for the velocities and support terms like "decreasing pattern" or "high-velocity zone". What are the velocities for these?

*We believe that, with the combination of the pairs of figures 4 and 6, and 5 and 7 (former numbers) these comments are now self-explanatory, as it can be easily visualized which stakes are faster and which are the corresponding velocity values.*

**P11/L11**: . . ., which has extremely "steep" slopes. (?)

*Done*

**P11/L21**: As mentioned before, I would not consider this as "root-mean-square position error".

*Removed mention to "root-mean-square".*

**P14/Appendix A**: EJ14 and others are missing in the table!

*The reason has been explained earlier. A mention to the loss of this particular stake has been added to Figure 3 caption: "The stake fell down to a newly opened frontal crevasse during 2010-2011 and was subsequently lost by iceberg calving, so it does not appear in Figure 2."*

Also I would avoid the error term in this table as long as it estimated based on formulae (7) and (8).

*We believe that the error term (which is now computed using the new Eq. (7) is the most significant piece of information in this table, as it gives an idea on the estimated positioning accuracies for the polynomial approximations to each stake trajectory. Consequently, this info has been kept, though the Table has now been moved to the Appendix (now Table B.1 in Appendix B) at the suggestion of the other reviewer.*

---

## Author Comment (AC2) · 2 Aug 2017

The authors present a 14-year dataset of in-situ measured surface velocities from an area in Antarctica that undergone a warming in the last decade(s). The dataset is very valuable for ice flow modelling or calibration of remotely sensed velocity fields (as already done by Otero et al., 2010; Osmanoglu et al., 2014). However, I have some comments that could help to improve the paper. As the other reviewer has already made many suggestions (that I have read, and I agree with most of them), I will try to avoid repeating suggestions.

**COMMENTS**

**Abstract**

I think, the abstract could state that velocities are only collected during summer season. So, possible seasonal variations are not covered.

> *The abstract already includes the sentence "**The measurements** cover the period 2000-2013 and **were done at the beginning and end of each austral summer season**." Consequently, from these data summer velocities as "extended winter" velocities can be calculated.*

**Introduction**

In general, I think the Introduction could better state the provided dataset was already used for tuning an ice flow model (Otero et al., 2010) and calibration of remotely sensed velocities (Osmanoglu et al., 2014)

> *We agree and have mentioned at the end of Section 1 that an earlier (and shorter) version of the presented dataset has already been used with such purposes, including the corresponding references.*

**Page1, Line 32**: It is correct, that observed velocities could be used as Dirichlet BC, but I would drop this sentence, because no ice sheet modeler apply that (to my knowledge).

> *We agree that **traditionally** glacier/ice sheet modellers have not used the velocities as Dirichlet BC. However, as mentioned later in the text (same paragraph) it is becoming more*

*and more common to solve an inverse Robin problem which involves two direct problems: one using Dirichlet BC at the glacier surface and another using Neumann BC at the glacier surface (and the misfit between both solutions is used to fit the viscosity and basal friction coefficients). Consequently, we have slightly modified the first sentence stating "**In theory,** they (observed surface velocities) could be directly used as Dirichlet boundary conditions … **However, the usual practice is** to set traction-free boundary conditions …" and later the more recent use of both Dirichlet BC is mentioned (as it was in the previous version of the manuscript).*

**Page 1, Line 34/37/40**: What do you mean with viscosity coefficient? In the viscosity relationships the ice hardness, effective strain hardness, a power coefficient, the effective strain rate and the enhancement factor appear. To my knowledge, the enhancement factor or the ice hardness is tuned.

*In our opinion, saying "viscosity coefficient" should be clear enough (at least for glacier (ice sheet modellers), in the sense that "coefficient" gives the idea of a multiplying factor (so powers are discarded) and also gives the idea of a constant (and the effective strain rate is a function of position). Of course it could be both ice hardness and/or enhancement factor (in fact, these can be joined together into a single coefficient or kept as two separate coefficients). The possible use of this coefficient as a function of position (as suggested in the procedure by Arthern and Gudmundsson (2010)) makes the interpretation of the term "viscosity coefficient" even more unclear. Consequently, we have added in brackets "ice hardness" to clarify the term.*

**Page 1, Line 34/37/40**: I prefer basal friction coefficient instead of basal drag coefficient.

*We agree and have changed it accordingly (3 occurrences).*

**Geographical Setting**

In **Fig 2**, I cannot identify the stake ID EJ14 (the one, that is plotted in Fig. 3). Also, a few more locations in Fig 2 are missing, if I compare the upper right inset in Fig. 3. The same holds for Fig. 5, it seems that some velocities at the marine terminating outlet of JG are not plotted (or not available?). I prefer to show every location in Fig. 2 and the following Figures. If in the result plots velocities are not available (for whatever reason), just show it with a black.

*Throughout the years some of the stakes have been lost (e.g. by iceberg calving at Johnsons Glacier front, or fallen down because of intensive melting, or buried by heavy snowfalls) and new ones have also been added (either as replacement, or to enlarge the available network of stakes). This is why stake EJ14 does not appear in Figure 2, as happens with some stakes appearing in later figures. In fact, what the figures show are snapshots in time which only include part of the stakes of the complete dataset in the PANGAEA database. To clarify this, we have added at the beginning of Section 3-Methods some clarifying sentences. In the particular case of stake EJ14, we have also noted in the caption of Figure 3 the fact that it was lost: "The stake fell down to a newly opened frontal crevasse during 2010-2011 and was subsequently lost by iceberg calving, so it does not*

*appear in Figure 2". With these clarifications, the suggestion by the reviewer of marking a black dot with the location of the missing stakes becomes unnecessary. Furthermore, this could be confusing. Note that the plots are snapshots in time and, for lost stakes, it would make no sense either to include its last recorded position (because it would correspond to an earlier time) or its "expected position" assuming that it had not been lost. The latter would be awkward in most cases, as many stakes have been lost by iceberg calving as they reach Johnsons front, so their "expected position" at the time of the snapshot would be at sea (at the proglacial bay).*

**Methods**

I think, the equations could be shortened by presenting only one component in each equation (similar as Eq. 6, 7, 8).

> *Indeed most equations could be shortened. In fact, at the request of the other reviewer, we have removed former equation (1), which was unnecessary, and rewritten former Eq. (2) (now Eq. (1)) in a simpler way. Also the subscript notation has been simplified/modified and in our opinion is now clearer. However, we believe that keeping the equations (1), (2) and (4) (with the new numbering; one unit more with the former numbering) for both X and Y coordinates makes the text more easily readable. Only equation (3) (formerly 4) could benefit of some shortening, but its location before Eq. (4) (which we believe should be kept in all space dimensions) would make awkward the use of a single space dimension, so we have preferred to maintain it as it was. We additionally note that Eq. (5) (formerly (6)) has also been simplified (some unneeded expressions removed) so the total "load" of equations is now lower (even if, at the request of the other reviewer, the equations regarding estimates for velocity errors -Eqs. (6) and following- have been modified and expanded).*

**Page 4, Line 1-4**: Can you explain the choice for the locations? Were you able to maintain the stakes for 14 years? So, none of the stakes was covered with snow or fallen down due to melting? However, I am a bit confused, as I count more than 60 stake locations in the provided shape file.

> *All of these aspects have been addressed in the first paragraph of the methods section of the modified manuscript:*
>
> *"The location of the stakes was chosen to provide a coverage as wide as possible of the entire glacier basins and their accumulation and ablation zones. Moreover, several sets of stakes were installed following glacier flowlines, thinking of possible glacier dynamics modelling studies. Ease of access for stake measurements and maintenance was also a consideration (e.g. some heavily crevassed areas had to be avoided, for safety reasons)."*
>
> *Regarding the number of stakes, we agree that the earlier writing of the manuscript was not clear enough, so we have clarified this aspect, again in the first paragraph of the new methods section (as well as other locations in the text, including some figure captions):*
>
> *"We note that, over time, some of the stakes have been lost (e.g. by iceberg calving at Johnsons Glacier front, or fallen down because of intensive melting, or buried by heavy*

*snowfalls) and new ones have also been added as replacement or to enlarge the original network. Because of this, there are differences in the set of stakes shown in the various figures in this paper, as they correspond to different snapshots in time. Also, the set of stakes included in the PANGAEA database (see Section 4) is larger than that in any of the figures, because it includes all of the stakes that have existed at any time within the complete measurement period."*

**Page 4, Line 5**: Could you explain, why you take the measurements several times in each season?

*At least one measurement at the beginning and another at the end of each summer season are performed. In this way, we are able to compute not only annual-averaged velocities but also summer velocities an "extended winter" (all year excluding the summer) velocities. In some cases a third measurement is done upon stake maintenance. Some years a further measurement was done in the middle of the summer to get a rough idea of the velocity evolution along the summer. Currently, just two measurements (at the start and end of the summer season are usually carried out). An important point that was not mentioned in the previous version of the manuscript is that Juan Carlos I station (which provides the logistic support for the measurements) is operated only during the austral summer, and this is why the measurements are limited to the summer period. These aspects have been clarified in the second paragraph of the new methods section.*

**Page 4, Line 6-13**: Could you give a few more details about the positioning accuracy? How have you determined the tilt at each stake? And how do you get an error estimate from the tilt?

*We have added some further info on the positioning accuracy (and in fact corrected some typos in the previous version). Over the years, the tilt of the stakes has been determined either using a clinometer (together with a compass, to measure the azimuth of the tilt) or by measuring by differential GNSS the coordinates of two points on the stake, to calculate from them the tilt and azimuth. The estimate of the error in tilt is a cumbersome process, depending on the particular measurement technique, not worth –in our opinion– to be described in the paper.*

As far as I understood, the positioning error does not enter the calculation of the velocity error (Eq. 7 and 8). The velocity error only depends on the polynomial interpolation. Is that right?
The positioning error should also be provided in the data repository for each stake. Please use same units (m or cm) when specifying the accuracy.

*You are partly right. Partly, because indeed it depends only on the polynomial approximation (current Eq. 4), but note that this one is derived from the polynomial approximation for positions (current Eq. 2), which is affected by the errors in position of the original stakes.*

*The new writing of current equations (6) and (7), and their associated text, clarifies the distinction between the errors in the observed velocities and their associated values in the polynomial function derived from the polynomial fit to the observed positions.*

Just curious: In the user manual of the Trimble 5700 receiver (https://www.ngs.noaa.gov/corbin/class_description/5700-5800V2UserGuide.pdf; Page 92), I found accuracies for both modes (RTK and faststatic) that are depending on the baseline length. Have you used these formulas?

> *No, we did not explicitly use such equation. In fact, what appears in page 92 is the minimum initialization time, and what we mention in the text is the occupation time (the initialization time was in fact larger, and dependent on measurement settings such as number of available (visible) satellites. Moreover, we note that there was a typo in this part of the text: the occupation time of 10 seconds actually corresponded to RTK measurements, while for fast-static measurements it was of 3-5 minutes:*
>
> *"The measurements were performed either in real-time kinematics (RTK) or in fast-static (post-processed) mode; for the former, an occupation time of 10 seconds was set, and for the latter it was of 3-5 minutes depending on the number of satellites available."*

**Page 4, Line 14**: Please rewrite. The procedure described here does not create the surface velocity map (-> nearest neighbor interpolation, Page 10, Line 1), it describes the time interpolation. You should also somewhere motivate the polynomial interpolation. I think, it is a nice method, but I don't really see the benefit by using the interpolated velocities compared to the direct measured velocities (just extrapolate the measured displacements to meter per year).

> *You are right. This has been modified. It now reads: "From the collected positions of the stakes at different epochs, the stake positions at any time can be estimated by applying the procedure described below." This, in fact, gives the main motivation, in the sense of providing a means for estimating the stake positions at any time from the polynomial interpolation of the observed positions. This is better than using interpolated/extrapolated displacements because it provides smoother particle trajectories.*

**Page 4, Eq. 3 and Table 1**: Can you specify the units of the coefficients a_x, b_x and c_x?

> *Done in the Table caption ($m\ y^{-2}$, $m\ y^{-1}$ and m). It is now Table B.1 in Appendix B.*

**Page 5, Eq. 6**: Sigma_x0^2 is not explained. In the legend the vector of residuals R is missing the subscript x.

> *Thanks for pointing this out. In the new version of the manuscript, the notation has been slightly changed, former eq. (6) (now Eq. 5) has been simplified and all terms are now explained.*

**Description of datasets**

In the shape file I found 26 fields instead of the 22 fields described here. I also suggest uploading the data for non-GIS users as a simple ascii file to the PANGAEA database. I think, the chapter could be moved to the Appendix.

> *The information has been added both in the description made in the text and in the files available in PANGAEA.*

*We have added a folder with the same information but in non-GIS users format (.xls and .txt).*
*We have also followed your suggestion regarding moving this section to an Appendix.*

**Results**

I recommend switching the Tables (Table 1 in the Appendix and the Table A1 in the main text), as the result of velocity calculation is the main task of this paper.

*We have followed your suggestion. You are fully right in that the main result to show in the paper is that on velocities, and the coefficients of the polynomial adjustment best fit in the Appendix (in fact, the most significant information in this table is that on the positioning errors).*

[Figure]

I plotted the data from the shape file in Qgis and have seen some strange behaviors (see attached figure). Most of the points are located on smooth lines/trajectories, but some outliers are observable. What is the reason for these outliers? Are these outliers considered in the velocity calculation or dropped?

*This is a subtle point. In fact, there are no outliers. Your plot induces to think about outliers or estrange behaviour because you are interpreting two different stakes (in each trajectory) as if they were a single one. In particular, it seems that your plot is representing the evolution over time of stakes EJ06 and EJ06r (which are different) and EJ14 and EJ14r (which are also different). At a certain point in history some stakes were installed at exactly the same position that was occupied some years earlier by another stake (if the original stake was named e.g. EJ14, the new stake installed in its former position was named EJ14r). And what you interpreting as strange behaviour is sample that the stake "r" is arriving at positions previously occupied by the stake without the "r" in its name). See figure below.*

[Figure]

*ADDITIONAL COMMENT: The complete story would be a much longer and complicated one: an earlier network of stakes, with fewer stakes, was deployed in the late 1990s in Johnsons Glacier –none in Hurd Glacier– by glaciologists from the University of Barcelona; they are the ones who did this experiment of installing new stakes in the former positions of another one, and in fact existed e.g. EJ14, EJ14r, EJ14rr, … This was an attempt to do some kind of "Eulerian" measurement of velocities (at fixed locations) versus the "Lagrangian" way of measurement, in which one follows the stakes as they move through the glacier.  However, this procedure revealed to be too cumbersome and time- and resource-demanding. When we "inherited" in 2000/01 this network of stakes, we deployed additional stakes on Johnsons Glacier and deployed a new network on Hurd Glaciers, and we discontinued this practice (of r, rr, … stakes) from our colleagues, though kept the "r" stakes until they were lost  (this took many years, and some are currently still "alive").*

*Of course we should not explain this story in the paper text, but we have remarked on the stake list in PANGAEA dataset that stakes such as EJ14 and EJ14r, etc. are actually different stakes.*

In order to reduce the number of figures, I think Figure 4,5 could be overlaid on Figure 6,7.

*We have followed your suggestion.*

**Figures and tables**

**Figure 3**: For colorblind people I recommend to use other colors than red and green.

*Done.*

**Figure 4**: The upper right inset marks JG instead of HG.

*Modified.*

**Figure 4,5**: I suggest enlarging the arrows or plotting the velocity magnitude with different colors.

*Done.*

**Figure 4-7**: I cannot read the stake IDs. Please enlarge the stake IDs or provide a better figure quality (pdf version looks fine, but my printout not).

*Done.*

**Table A1**: The error provided is derived from Eq. 8?

*Yes, but in the current version of the manuscript the corresponding data are given in Table 1 and the velocity errors derived from Eq. (7).*

For all tables, please move the caption to the top.

*Done.*

---

## Author Comment (AC3) · 5 Aug 2017

[revised manuscript text omitted]

The Hurd Peninsula ice cap is subjected to the maritime climate of the western Antarctic Peninsula (AP) region. The annual average temperature at JCI during the period 1994-2014 was −1.2ºC, with average summer (DJF) and winter (JJA) temperatures of 1.9ºC and −4.7ºC, respectively (Bañón and Vasallo, 2016). Summer, winter and annual mass balances have been measured using the glaciological method on the same network of stakes used for the glacier velocity measurements, and then integrated to the entire glacier basins. The mean surface mass balances over the period 2002-2011 have not been significantly different from zero for either glacier: 0.05±0.30 m w.e. for Johnsons and −0.15±0.44 m w.e. for Hurd. The ranges indicate the standard deviations, showing that the mass balances have a noticeable interannual variability. The estimated errors of the annual mass balances are lower, of the order of ±0.1 m w.e. The slightly more negative balance for Hurd Glacier is due to its lower accumulation rates, attributed to snow redistribution by wind, together with slightly higher ablation rates, due to Hurd's hypsometry, which shows a much larger share of area at the lowermost altitudes (<100 m) as compared with Johnsons (Navarro et al., 2013). The average accumulation area ratios over the same period were 44±24 % for Hurd Glacier and 61±21 % for Johnsons Glacier (again, quoted the standard deviations). Their equilibrium line altitudes (ELA) for the same period were 228±57 m a.s.l. and 187±37 m a.s.l., respectively (Navarro et al., 2013).

[Figure]

*Fig. 2. Network of stakes on Hurd and Johnsons Glaciers at the end of the 2012-2013 Antarctic summer campaign. (Base map: SGE, 1990).*

**3. Methods**

The glacier surface velocities were estimated based on repeated differential GNSS measurements in a network of stakes deployed by the authors on Johnsons and Hurd glaciers. The network of stakes consisted (at the end of the measurement period reported) of 22 stakes for Johnsons and 30 stakes for Hurd Glacier (Fig. 2). The location of the stakes was chosen to provide a coverage as wide as possible of the entire glacier basins and their accumulation and ablation zones. Moreover, several sets of stakes were installed following glacier flowlines, thinking of possible glacier dynamics modelling studies. Ease of access for stake measurements and maintenance was also a consideration (e.g. some heavily crevassed areas had to be avoided, for safety reasons). We note that, over time, some of the stakes have been lost (e.g. by iceberg calving at Johnsons Glacier front, or fallen because of intensive melting, or buried by heavy snowfalls) and new ones have also been added as replacement or to enlarge the original network. Because of this, there are differences in the set of stakes shown in the various figures in this paper, as they correspond to different snapshots in time. Also, the set of stakes included in the PANGAEA database (see Section 4) is larger than that in any of the figures, because it includes all of the stakes that have existed at any time within the complete measurement period.

The stakes were surveyed 2-4 times per summer campaign during the period 2000-2013. Measurements are restricted to the summer season because Juan Carlos I station (which provides the logistic support for fieldwork) is operated only during the austral summer. At least one measurement at the beginning and another at the end of each summer season are performed. In this way, we are able to compute not only annual-averaged velocities but also summer velocities and "extended winter" (all year excluding the summer) velocities. In some cases additional measurements are carried out during the summer in order to get a rough idea of the temporal variations in velocity along the summer. The GNSS measurements were carried out using a Trimble 5700 system, with data controller TSC2. The measurements were performed either in real-time kinematics (RTK) or in fast-static (post-processed) mode; for the former, an occupation time of 10 seconds was set, and for the latter it was of 3-5 minutes depending on the number of satellites available. In general, RTK mode was used, but in some cases a radio link to the base station was not available, and fast-static mode was employed. The GNSS base station was located at the neighbouring Juan Carlos I Station (Fig. 1), at a distance within 2-4 km from each stake measurement point. The base station Juan Carlos I is a permanent GNSS station with coordinates determined with an accuracy better than 0.007 m in horizontal and 0.012 m in vertical (Ramírez-Rodríguez, 2007). The estimated horizontal accuracy for the stake positions lies between 0.07 and 0.60 m. The main contributor to this uncertainty is not the GNSS measurement error (which has average values of 0.07 and 0.10 m for horizontal and vertical positioning, respectively) but the estimated uncertainties in the correction for tilt of the stakes. The estimated coordinates of the stakes were projected into the UTM-System for Zone 20S.

From the collected positions of the stakes at different epochs, the stake positions at any time can be estimated by applying the procedure described below. We will just focus on horizontal velocities, since the vertical component of the velocity is very small, and prone to errors such as those of tilt of the stake. From the known position $(x_{t_i}, y_{t_i})$ of a stake at a given time $t_i$ (expressed in days since the zero time for observations, $t_0$, which we arbitrarily set as 01/01/1999 at 00:00), with the subscript $i$ indicating the sequential number of the observation (from $i = 1$ to $i = n$), we define the planimetric position of a stake over time (i.e. its trajectory) by the discrete functions

$$X(t_i) = X(x_{t_1}, x_{t_2}, \ldots, x_{t_n})$$
$$Y(t_i) = Y(y_{t_1}, y_{t_2}, \ldots, y_{t_n})$$
(1)

It is possible to adjust the previous functions by means of second-order polynomials, which is equivalent to assuming that the stake moves with constant acceleration:

$$X_a(t_i) = a_x t_i^2 + b_x t_i + c_x$$
$$Y_a(t_i) = a_y t_i^2 + b_y t_i + c_y$$
(2)

This set of two equations, with three unknowns each, will have a solution, or a better approximation to it, if sufficient observations ($n \geq 3$) are available for each stake. The unknown coefficients are determined by the least-square fitting method, minimizing the residual vectors

$$\mathbf{R}_x = \begin{bmatrix} t_1^2 & t_1 & 1 \\ t_2^2 & t_2 & 1 \\ \ldots & \ldots & \ldots \\ t_n^2 & t_n & 1 \end{bmatrix} \begin{bmatrix} a_x \\ b_x \\ c_x \end{bmatrix} - \begin{bmatrix} X(t_1) \\ X(t_2) \\ \ldots \\ X(t_n) \end{bmatrix} = \mathbf{AC}_x - \mathbf{X}$$

.  (3)

$$\mathbf{R}_y = \begin{bmatrix} t_1^2 & t_1 & 1 \\ t_2^2 & t_2 & 1 \\ \ldots & \ldots & \ldots \\ t_n^2 & t_n & 1 \end{bmatrix} \begin{bmatrix} a_y \\ b_y \\ c_y \end{bmatrix} - \begin{bmatrix} Y(t_1) \\ Y(t_2) \\ \ldots \\ Y(t_n) \end{bmatrix} = \mathbf{AC}_y - \mathbf{Y}$$

By minimizing the above residuals for each of the existing stakes, we will get the adjustment functions, $X_a(t_i)$ and $Y_a(t_i)$, which allow to estimate how the position of each stake evolves with time.

The horizontal velocity of a stake will be given, from the time derivatives of the positions, by the expressions:

$$\mathbf{v} = v_x\mathbf{i} + v_y\mathbf{j}$$
$$v_x = X_a'(t_i) = 2\,a_x\,t_i + b_x$$
$$v_y = Y_a'(t_i) = 2\,a_y\,t_i + b_y \tag{4}$$
$$v_{xy} = \sqrt{v_x^2 + v_y^2}$$

To obtain the error estimates $e_x$ and $e_y$ of the adjusted functions, $X_a(t_i)$ and $Y_a(t_i)$ (from which we calculate the error in horizontal positioning as $e_{xy} = \sqrt{e_x^2 + e_y^2}$) we follow the parametric adjustment procedure (see details in Ghilani, 2010), which has to be applied separately for $X$ and $Y$ (for brevity, we just describe it below for $X$). For a least square approximation, assuming observations of equal weight, these equations are:

$$\mathbf{N} = [\mathbf{A}^{\mathrm{T}}\,\mathbf{A}]$$

$$\mathbf{C}_x = \mathbf{N}^{-1}\,[\mathbf{A}^{\mathrm{T}}\,\mathbf{X}]$$

$$\mathbf{R}_x = \mathbf{A}\,\mathbf{C}_x - \mathbf{X}$$

$$\widetilde{\mathbf{X}} = \mathbf{X} + \mathbf{R}_x$$

$$e_x = \sqrt{\dfrac{\mathbf{R}_x^{\mathrm{T}}\mathbf{R}_x}{r}}$$

**X**: Vector of observations

$\widetilde{\mathbf{X}}$: Vector of estimates

**A**: Matrix of coefficients

$\mathbf{R}_x$: Vector of residuals

$\mathbf{C}_x$: Vector of unknowns (the coefficients in the polynomial adjustment)

**N**: Cost or discrepancy matrix

$e_x^2$: Reference variance

$r$: Number of degrees of freedom; $r = n - 3$ , with $n$ the number of observations

(5)

The above equations are solved for each individual stake. $\mathbf{C}_x$ is solved first to determine the coefficients of the second-degree polynomial adjustment. Then, the adjusted values $\mathbf{AC}_x$ are calculated and the residuals $\mathbf{R}_x$ computed, and finally the error in position $e_x$ is calculated. The process is repeated for the corresponding equations for the Y coordinate, to get the vector of polynomial adjustment coefficients $\mathbf{C}_y$ and the error estimate $e_y$.

We note that the above error estimates do not represent actual errors in the data points but the standard deviations of the data point positions with respect to their corresponding values (for the same time $t$) in the polynomial approximation defined by Equations (2).

If we were to estimate the error in velocity for a given stake between two particular positions $\mathbf{x}_i = (x_i, y_i)$, $\mathbf{x}_{i+1} = (x_{i+1}, y_{i+1})$, with positioning errors $e_{\mathbf{x}_i}$, $e_{\mathbf{x}_{i+1}}$, respectively, separated by a time interval $\Delta t = t_{i+1} - t_i$ (i.e. the error in $\mathbf{v}_i = \frac{\mathbf{x}_{i+1}-\mathbf{x}_i}{\Delta t}$), it would be given by

$$e_{\mathbf{v}_i} = \frac{1}{\Delta t}\sqrt{e_{\mathbf{x}_i}^2 + e_{\mathbf{x}_{i+1}}^2} \tag{6}$$

However, if we are interested in the error in the velocity given by the polynomial approximation defined by Equation (4), this would be given by the root-mean-square error of the deviations between the average velocities calculated for each time interval $\mathbf{v}_i^{obs} = \frac{\mathbf{x}_{i+1}-\mathbf{x}_i}{\Delta t} = (v_{x_i}^{obs}, v_{y_i}^{obs})$ and the corresponding velocities $\mathbf{v}_i^{pol} = (v_{x_i}^{pol}, v_{y_i}^{pol})$ calculated using equation (4) for time $\frac{t_i+t_{i+1}}{2}$, i.e.

$$e_{v_x} = \sqrt{\frac{1}{N}\sum_{i=1}^{N}(v_{x_i}^{obs} - v_{x_i}^{pol})^2}$$

$$e_{v_y} = \sqrt{\frac{1}{N}\sum_{i=1}^{N}(v_{y_i}^{obs} - v_{y_i}^{pol})^2} \tag{7}$$

$$e_{v_{xy}} = \sqrt{e_{v_x}^2 + e_{v_y}^2}$$

where the superscripts *obs* and *pol* denote observed (calculated from observations) and polynomial (calculated using the polynomial approximation (4)) values, respectively, and $N$ represents the number of velocity intervals ($N = n - 1$, with $n$ the number of stake position observations). Note that the value given by Equation (7) is a single value

representing the average error for each polynomial approximation (i.e. a single error value for each stake), while the errors given by Equation (6) are interval velocity errors between two consecutive positions of a given stake.

**4. Results**

5 The procedure described in the above section was applied to every stake that has existed for any subperiod (perhaps the entire period) within the complete measurement period 2000-2013. The whole data for the stakes is available at PANGAEA database (http://doi.pangaea.de/ 10.1594/PANGAEA.846791) and is described in Appendix A.

The results for the coefficients of the polynomial adjustments for the stake positions, and the estimated horizontal positioning misfits for each stake, are given in Table B.1 of Appendix B. To give an idea of the order of magnitude of 10 the velocities and their associated errors, as well as their spatial variations, we have included in Table 1 the calculated values for a given time.

*Table 1. Horizontal velocities for Hurd and Johnsons Glacier stakes on 13/02/2013, calculated using the first-degree polynomial for velocity derived from the second-degree polynomial adjustment for stake positions. From left to right: stake name, X and Y components of horizontal velocity ($v_x$, $v_y$), absolute value of horizontal velocity ($v_{xy}$), azimuth of* 15 *the horizontal velocity vector (θ), and error estimate for the horizontal velocity ($e_{v_{xy}}$), calculated using Equation (7). All velocities are expressed in meters per year.*

| Stake | $v_x$ (m y$^{-1}$) | $v_y$ (m y$^{-1}$) | $v_{xy}$ (m y$^{-1}$) | θ (°) | $e_{v_{xy}}$ (m y$^{-1}$) |
|---|---|---|---|---|---|
| EH01 | -0,35 | -0,61 | 0,70 | 210,09 | ±0,40 |
| EH02 | -0,81 | -0,02 | 0,81 | 268,62 | ±0,45 |
| EH03 | -0,88 | -0,34 | 0,95 | 248,93 | ±0,44 |
| EH04 | -1,69 | -0,44 | 1,75 | 255,46 | ±0,28 |
| EH05 | -1,94 | -0,81 | 2,10 | 247,45 | ±0,19 |
| EH06 | -2,91 | -1,08 | 3,10 | 249,64 | ±0,26 |
| EH07 | -2,83 | -1,78 | 3,35 | 237,85 | ±0,71 |
| EH08 | -2,27 | -1,89 | 2,95 | 230,20 | ±0,29 |
| EH10 | -0,85 | 0,98 | 1,30 | 319,35 | ±0,19 |
| EH11 | -1,24 | 1,41 | 1,87 | 318,75 | ±0,24 |
| EH12 | -1,04 | 0,54 | 1,17 | 297,42 | ±1,61 |
| EH13 | -1,12 | 1,52 | 1,89 | 323,61 | ±0,22 |
| EH14 | -0,93 | 0,34 | 0,99 | 290,06 | ±0,78 |
| EH19 | -1,53 | 0,49 | 1,60 | 287,67 | ±0,28 |
| EH20 | -1,53 | -0,58 | 1,64 | 249,14 | ±0,57 |
| EH21 | -2,27 | -9,78 | 10,04 | 193,08 | ±0,59 |
| EH22 | -0,56 | -0,78 | 0,96 | 215,99 | ±0,73 |
| EH25 | -2,08 | -0,08 | 2,08 | 267,85 | ±0,16 |
| EH27 | -2,48 | -1,31 | 2,81 | 242,27 | ±0,23 |
| EH28 | -1,26 | -0,56 | 1,38 | 245,91 | ±0,39 |
| EH30 | -1,87 | -2,84 | 3,40 | 213,30 | ±4,90 |
| EH31 | -1,10 | -0,64 | 1,27 | 239,86 | ±0,85 |
| EH32 | -0,48 | 0,28 | 0,56 | 300,13 | ±1,03 |
| EH35 | -2,15 | 0,77 | 2,29 | 289,76 | ±0,47 |
| EH36 | -1,66 | 1,83 | 2,47 | 317,76 | ±1,08 |
| EH37 | -2,28 | 1,61 | 2,79 | 305,17 | ±1,37 |
| EH38 | -2,19 | -1,93 | 2,92 | 228,65 | ±0,65 |
| EH39 | -2,22 | -1,55 | 2,71 | 235,15 | ±0,41 |
| EH40 | -2,20 | 2,08 | 3,03 | 313,33 | ±0,30 |
| EH41 | -0,66 | -0,71 | 0,97 | 223,07 | ±1,71 |

| Stake | $v_x$ (m y$^{-1}$) | $v_y$ (m y$^{-1}$) | $v_{xy}$ (m y$^{-1}$) | $\theta$ (°) | $e_{v_{xy}}$ (m y$^{-1}$) |
|---|---|---|---|---|---|
| EJ03 | 2,42 | 6,57 | 7,00 | 20,25 | ±0,66 |
| EJ04 | 0,85 | 7,28 | 7,33 | 6,64 | ±0,96 |
| EJ05 | 0,61 | 10,94 | 10,95 | 3,17 | ±1,92 |
| EJ06 | -5,73 | 23,50 | 24,18 | 346,30 | ±0,47 |
| EJ09 | 0,01 | -0,01 | 0,02 | 135,41 | ±0,33 |
| EJ10 | -4,11 | -1,95 | 4,55 | 244,64 | ±1,07 |
| EJ16 | -7,41 | 13,95 | 15,80 | 332,01 | ±2,29 |
| EJ18 | -22,56 | 29,31 | 36,99 | 322,42 | ±0,17 |
| EJ21 | -0,63 | 1,01 | 1,19 | 327,85 | ±0,42 |
| EJ22 | -1,36 | 3,56 | 3,81 | 339,08 | ±0,62 |
| EJ23 | -1,82 | 6,53 | 6,78 | 344,39 | ±0,31 |
| EJ24 | 0,99 | 5,15 | 5,25 | 10,93 | ±0,58 |
| EJ26 | -7,18 | -2,68 | 7,67 | 249,55 | ±1,19 |
| EJ27 | -13,50 | -2,37 | 13,70 | 260,03 | ±0,29 |
| EJ29 | 3,53 | 3,24 | 4,79 | 47,47 | ±0,08 |
| EJ30 | -1,87 | -1,56 | 2,43 | 230,24 | ±0,16 |
| EJ31 | 1,36 | 2,73 | 3,05 | 26,45 | ±0,39 |
| EJ32 | 1,96 | 2,22 | 2,96 | 41,46 | ±1,83 |
| EJ33 | -14,23 | 5,57 | 15,28 | 291,38 | ±0,33 |
| EJ34 | 0,21 | 1,63 | 1,64 | 7,45 | ±7,14 |
| EJ35 | -6,29 | -0,41 | 6,31 | 266,30 | ±0,40 |

[Figure]

*Figure 3. Map showing the time evolution of Stake EJ14. Horizontal velocities and times for various positions are shown. The stake fell down to a newly opened frontal crevasse during 2010-2011 and was subsequently lost by iceberg calving, so it does not appear in Figure 2. The inset to the right shows the location of the image shown to the left (in the inset, EJ14 positions are shown in green). In this, and the following figures, UTM coordinates (sheet 20S) are indicated. The background image is a satellite photo of the QUICKBIRD system program (2007).*

As an example, the detailed results for a particular stake, EJ14, are shown in Table 2 and Figure 3. The latter shows the position changes of the stake over time.

*Table 2. Example of results for the adjustment by least squares of the position and the velocity of a stake (EJ14, near the calving front of Johnsons Glacier; see Figure 3), together with the deviations from the polynomial approximation for the position, as well as the maximum horizontal velocity and its direction.*

$$X_a(t_i) = -0.0000083181\, t_i{}^2 + 0.0057260572\, t_i + 635350.340$$
$$Y_a(t_i) = 0.0000190604\, t_i{}^2 - 0.0112107159\, t_i + 3048898.260$$
$$v_x = -0.0000166362\, t_i + 0.0057260572$$
$$v_y = 0.0000381208\, t_i - 0.0112107159$$
$$e_x = \pm1.69\, m$$
$$e_y = \pm4.46\, m$$
$$e_{xy} = \pm4.77\, m$$
$$n = 25$$
$$Maximum\ velocity: 57.31\ m\ y^{-1}\ on\ March\ 1, 2010.$$
$$Maximum\ velocity\ azimuth: 336.7019°$$

[Figure]

*Figure 4. Map of contour lines of the absolute values of the horizontal velocity, for Hurd Glacier, obtained from the spatial interpolation of the corresponding vector velocity field, calculated for 13/02/2013 using the first-degree polynomial derived from the second-degree polynomial adjustment of the stake positions. The numerical values for the absolute value of velocity at each stakes (in brackets) and the vector velocity directions (arrows) are also represented. The yellow near-zero velocity band indicates the approximate location of the ice divides.*

In Figures 4 and 5 we show the horizontal velocities (absolute values and directions) for all stakes of Hurd and Johnsons glaciers, respectively, for a given date (13/02/2013), calculated using the corresponding polynomial adjustments. We also show the corresponding contour lines of the absolute value of the velocities for the same date, calculated from the

spatial interpolation of the velocity vector field. Maximum velocities on Hurd Glacier are only of a few m y⁻¹, and approach 10 m y⁻¹ at the head of the unnamed glacier draining towards the south. Maximum velocities on Johnsons Glacier are much larger, up to several tens of m y⁻¹, and reached 65 m y⁻¹ near the calving front. The location of the main ice divides is apparent in the contour plots (zero velocity bands).

[Figure]

*Figure 5. Map of contour lines of the absolute values of the horizontal velocity, for Johnsons Glacier, obtained from the spatial interpolation of the corresponding vector velocity field, calculated for 13/02/2013 using the first-degree polynomial derived from the second-degree polynomial adjustment of the stake positions. The numerical values for the absolute value of velocity at each stakes (in brackets) and the vector velocity directions (arrows) are also represented. The yellow near-zero velocity band indicates the approximate location of the ice divides (except for the zone to the east, between UTM northing 3048000 and 3048500, which corresponds to thin frozen-to-bed ice on the upper part of a nunatak).*

**5. Discussion and summarizing conclusions**

[revised manuscript text omitted]

Another shortcoming of the presented dataset is that it does not allow for an easy analysis of dynamical response to climate changes (such as those regionally observed by Oliva et al., 2016), because what is available is a Lagrangian velocity field (velocities measured at stakes that change their position with time), while what is needed for studying glacier velocity variations in response to climate changes is an Eulerian velocity field (velocities measured at fixed location in space).

From the above discussion, a desirable complement to the available in situ velocity dataset presented in this paper would be a continuous record of ice velocities at selected stakes.

Summarizing, the presented dataset is a useful source of input data for numerical models of glacier dynamics and for calibration-validation of remotely-sensed velocity data. It fills an observational data gap in the region peripheral to the Antarctic Peninsula, and it is thus expected that these data will contribute to the understanding of the dynamics of the ice masses in this region and their response to environmental changes.

**Data availability**

[revised manuscript text omitted]

**Appendix A**

The shape file CNDA-ESP_SIMRAD_VELOCITY.shp available at PANGAEA database (http://doi.pangaea.de/ 10.1594/PANGAEA.846791), and its corresponding versions in Excel (.xlsx) and ASCII (.txt) formats, contain the position data for all stakes of Johnsons and Hurd glaciers for the period from 2000 to 2013. We describe below the contents of each individual field in the shape file, as described in file "fields_explanation.txt". We remind the reader that the set of stakes included in the data files is larger than that shown in the various figures in this paper, as it includes all stakes that have existed for any period within the entire measurement period, while the figures give snapshots in time. The PANGAEA data files also include a table (file "stake_dates.txt") indicating the dates of the start and the end of the measurement period for each stake.

- Field "t38_stake": The name of the stake under consideration (see stakes in Figure 2).
- Field "t38_t0": The zero time for the time variable. We set it as 01/01/1999 at 00:00 GMT.
- Field "t38_fecha": The date and time for the measurement, with "YYYYMMDDHHMMSS" format.
- Field "t38_inct": The period of time in days from "t38_t0" to "t38_fecha" ($t_n$ in the above equations).
- Field "t38_x": X coordinate in meters (UTM 20S) for the stake (considered in an ideal vertical position, after correction for tilt, if applicable) ($x_{t_n}$ in Equation 1).
- Field "t38_y": Y coordinate in meters (UTM 20S) for the stake (considered in an ideal vertical position, after correction for tilt, if applicable) ($y_{t_n}$ in Equation 1).
- Field "t38_x_ide": X coordinate in meters (UTM 20S) for the position of the stake for the given time, calculated using the second-degree polynomial adjustment ($X_a(t_n)$.in Equation 2).
- Field "t38_y_ide": Y coordinate in meters (UTM 20S) for the position of the stake for the given time, calculated using the second-degree polynomial adjustment ($Y_a(t_n)$ in Equation 2).
- Field "t38_vx": X component for horizontal velocity of the stake for the given time, expressed in meters per year, calculated from the second-degree polynomial adjustment ($v_x$ in Equation 4).
- Field "t38_vy": Y component for horizontal velocity of the stake for the given time, expressed in meters per year, calculated from the second-degree polynomial adjustment ($v_y$ in Equation 4).
- Field "t38_vxy": Absolute value of horizontal velocity of the stake for the given time, expressed in meters per year, calculated from the X and Y components of the velocity obtained from the second-degree polynomial adjustment ($v_{xy}$ in Equation 4).
- Field "t38_v_aci": Azimuth for horizontal velocity of the stake, expressed in sexagesimal degrees, at the date of the measurement.
- Field "t38_err_x": Root-mean-squared deviation for the X position of the stake, expressed in meters ($e_x$).
- Field "t38_err_y": Root-mean-squared deviation for the Y position of the stake, expressed in meters ($e_y$).
- Field "t38_max_x": Maximum error obtained for the X position of the stake, expressed in meters.
- Field "t38_max_y": Maximum error obtained for the Y position of the stake, expressed in meters.
- Field "t38_ax": The estimation for the "$a_x$" coefficient in the second-degree polynomial adjustment of the position X of the stake ($a_x$ in Equation 2).
- Field "t38_bx": The estimation for the "$b_x$" coefficient in the second-degree polynomial adjustment of the position X of the stake ($b_x$ in Equation 2).
- Field "t38_cx": The estimation for the "$c_x$" coefficient in the second-degree polynomial adjustment of the position X of the stake ($c_x$ in Equation 2).
- Field "t38_ay": The estimation for the "$a_y$" coefficient in the second-degree polynomial adjustment of the position Y of the stake ($a_y$ in Equation 2).
- Field "t38_by": The estimation for the "$b_y$" coefficient in the second-degree polynomial adjustment of the position Y of the stake ($b_y$ in Equation 2).
- Field "t38_cy": The estimation for the "$c_y$" coefficient in the second-degree polynomial adjustment of the position Y of the stake ($c_y$ in Equation 2).
- Field "dias": Days after t38_t0 for a simulation. In this example, 5817 days.
- Field "prevista_x": Example of X coordinate in meters (UTM 20S) for the stake (considered in an ideal vertical position, after correction for tilt) after 5817 days.
- Field "prevista_y": Example of Y coordinate in meters (UTM 20S) for the stake (considered in an ideal vertical position, after correction for tilt) after 5817 days.
- Field "movxy": Planimetric movement in meters (UTM 20S) for the stake (considered in an ideal vertical position, after correction for tilt) after 5817 days.

Table B.1. Polynomial coefficients of the adjustment functions, $X_a(t_i)$ and $Y_a(t_i)$, according to equation (2) for all the stakes of the glaciers under study. The units for the coefficients a, b and c are m y$^{-2}$, m y$^{-1}$ and m, respectively. The Table also shows the estimated horizontal position errors $e_{xy} = \sqrt{e_x^2 + e_y^2}$) (in meters) involved in the polynomial approximation of the position.

[revised manuscript text omitted]

---

## Author Comment (AC4) · 5 Aug 2017

In order to make our comments clearer and following the editorial recommendations we include a marked-up version of the manuscript in which we have highlighted the main modifications made due to both reviewers.

Please also note the supplement to this comment: https://www.earth-syst-sci-data-discuss.net/essd-2017-30/essd-2017-30-AC4-supplement.pdf